# H-FEX: A Symbolic Learning Method for Hamiltonian Systems

## Abstract

Hamiltonian systems describe a broad class of dynamical systems governed by Hamiltonian functions, which encode the total energy and dictate the evolution of the system. Data-driven approaches, such as symbolic regression and neural network-based methods, provide a means to learn the governing equations of dynamical systems directly from observational data of Hamiltonian systems. However, these methods often struggle to accurately capture complex Hamiltonian functions while preserving energy conservation. To overcome this limitation, we propose the Finite Expression Method for learning Hamiltonian Systems (H-FEX), a symbolic learning method that introduces novel interaction nodes designed to capture intricate interaction terms effectively. Our experiments, including those on highly stiff dynamical systems, demonstrate that H-FEX can recover Hamiltonian functions of complex systems that accurately capture system dynamics and preserve energy over long time horizons. These findings highlight the potential of H-FEX as a powerful framework for discovering closed-form expressions of complex dynamical systems.

## 1 Introduction

Symbolic regression is a technique for discovering mathematical equations from data, making it a useful tool for recovering the underlying laws of physical systems (Brunton et al., 2016). Unlike traditional regression, which fits parameters to a predefined functional form, symbolic regression searches the space of mathematical expressions to find the model that best fits the dataset. One important class of dynamical systems where such methods are highly applicable is Hamiltonian systems (DiPietro et al., 2020; DiPietro & Zhu, 2022), which describe a wide range of physical phenomena, including celestial mechanics (Aarseth, 2003), quantum mechanics (Dickey, 2003), and control systems (Khan & Storkey, 2022). These systems are governed by Hamiltonian functions, which encode the total energy of the system and determine its evolution through Hamilton's equations. A key aspect of modeling Hamiltonian systems is energy conservation (Bilbao et al., 2023), which is essential for ensuring physically meaningful behavior, especially over long time horizons. Given their importance, Hamiltonian systems are a natural setting for symbolic regression methods, which can discover Hamiltonian functions directly from data.

While there is a need for data-driven discovery of Hamiltonian functions, existing methods struggle to balance interpretability with adherence to energy conservation laws, particularly when dealing with the complexities of dynamical systems. General symbolic regression methods such as AI Feynman (Udrescu & Tegmark, 2020), physical symbolic optimization ($\Phi$-SO) (Tenachi et al., 2023), and incontext symbolic regression (ICSR) (Merler et al., 2024) are not natively designed to learn dynamics directly from trajectory data, nor can they enforce Hamiltonian constraints or jointly fit multiple coupled output equations. Even methods tailored to ODE systems such as ODEFormer (d'Ascoli et al., 2023) can face other difficulties such as handling multiple trajectory datasets. Another well-known symbolic regression method for discovering governing equations from data is sparse-identification of nonlinear dynamics (SINDy) (Brunton et al., 2016), which recovers equations by selecting terms from a predefined set of basis functions. However, on its own, SINDy does not enforce Hamiltonian dynamics, and its reliance on linear combinations of basis functions limits its ability to represent complex Hamiltonian functions. If we move from symbolic methods to neural network methods, Hamiltonian Neural Networks (HNNs) (Greydanus et al., 2019) explicitly enforce Hamiltonian

dynamics by modeling a surrogate of the Hamiltonian function, ensuring energy conservation. Despite this advantage, HNNs lack interpretability, as their learned representations are encoded in black-box neural networks. More advanced variants, such as Symplectic Recurrent Neural Networks (SRNNs) (Chen et al., 2019), which incorporate multi-step integration, and Stiffness-Aware Neural Networks (SANNs) (Liang et al., 2021), which can identify stiff portions of training data, improve robustness and accuracy. However, they still rely on neural networks, which makes it difficult to produce meaningful closed-form expressions and to ensure accurate long-term predictions. These limitations highlight the need for a symbolic regression method tailored for Hamiltonian systems, one that can learn complex Hamiltonian functions accurately while respecting conservation laws.

Finite Expression Method (FEX) is a symbolic regression method that leverages reinforcement learning to discover complex closed-form expressions from data (Jiang et al., 2023; Liang & Yang, 2022; Song et al., 2024) using reinforcement learning. Unlike SINDy, which relies on linear combinations of predefined basis functions, FEX represents expressions as trees of operators with associated weights. This structure enables it to capture a broader class of mathematical expressions, including those involving function composition. However, FEX is not designed to explicitly enforce Hamiltonian dynamics, making a direct application to Hamiltonian systems challenging. Additionally, many Hamiltonian systems (Gowers, 2008; Makarov, 2018; Wu et al., 2025) contain interaction terms that represent dependencies between multiple interacting objects. FEX lacks a way to model these interaction terms, further limiting its ability to model complex Hamiltonian systems.

To address the shortcomings, we introduce Finite Expression Method for Hamiltonian Systems (H-FEX), an adaptation of FEX specifically designed for Hamiltonian systems. H-FEX modifies the search loop of FEX to enforce Hamiltonian dynamics, similar to how HNNs impose Hamiltonian constraints to respect conservation laws. Moreover, we introduce interaction nodes, which enable the construction of interaction terms that capture dependencies between interacting objects. Through numerical experiments, we demonstrate that H-FEX accurately recovers interpretable, closed-form representations of the Hamiltonian function. Additionally, H-FEX outperforms other methods by generating trajectories that are highly accurate and preserve energy over long time horizons. By bridging the gap between FEX and Hamiltonian systems, H-FEX is a powerful tool for discovering closed-form representations of Hamiltonian systems directly from data.

The remainder of this paper is organized as follows. Section 2 introduces Hamiltonian systems and defines the loss function used to train a Hamiltonian surrogate. Section 3 provides an overview of H-FEX, describes the interaction nodes, and explains how integrators are used during training and evaluation. Section 4 presents experiments benchmarking H-FEX against existing methods. Finally, Section 6 summarizes the contributions and outlines directions for future research.

## 2 Problem Statement

A Hamiltonian system is a dynamical system characterized by the Hamiltonian function $\mathcal{H}(\mathbf{p}, \mathbf{q})$, which is a real-valued function $\mathcal{H} : \mathbb{R}^{2d} \to \mathbb{R}$ that maps the momentum $\mathbf{p} \in \mathbb{R}^d$ and the position $\mathbf{q} \in \mathbb{R}^d$ to a scalar energy. A Hamiltonian system is said to be separable if it can be written as $\mathcal{H}(\mathbf{p}, \mathbf{q}) = K(\mathbf{p}) + U(\mathbf{q})$, where $K$ and $U$ denote the kinectic and potential energy, respectively. The phase space $\mathbb{R}^{2d}$ represents the space of all possible states in the system. The time evolution of the system is governed by Hamilton's equations:

$$\frac{d\mathbf{p}}{dt} = -\frac{\partial \mathcal{H}}{\partial \mathbf{q}}, \quad \frac{d\mathbf{q}}{dt} = \frac{\partial \mathcal{H}}{\partial \mathbf{p}}. \tag{1}$$

Trajectories in a Hamiltonian system describe how a particle evolves over time, starting from an initial condition.

Integrators are numerical methods that compute trajectories over a time interval, given an initial condition (Hairer et al., 1993). Consider the discretization of an interval $[0, T]$ using uniformly spaced time points $(t_n)_{n=0}^N$, where $t_0 = 0$ and $t_N = T$. Given a Hamiltonian function $\mathcal{H}$ and an initial condition $(\mathbf{p}_{t_0}, \mathbf{q}_{t_0})$, an integrator computes the trajectory evaluated at time points $(t_n)_{n=0}^N$:

$$(\mathbf{p}_{t_n}, \mathbf{q}_{t_n})_{n=0}^N = \text{Integrator}(\mathcal{H}, (\mathbf{p}_{t_0}, \mathbf{q}_{t_0}), (t_n)_{n=0}^N). \tag{2}$$

Given observed trajectories $(\mathbf{p}_{t_n}, \mathbf{q}_{t_n})_{n=0}^N$ from an unknown Hamiltonian system, where the initial condition $(\mathbf{p}_{t_0}, \mathbf{q}_{t_0})$ lies in a domain $\Omega \subseteq \mathbb{R}^{2d}$, the goal is to learn a parameterized surrogate Hamiltonian $\hat{\mathcal{H}}_{\boldsymbol{\Theta}}$ that approximates the true, unknown Hamiltonian $\mathcal{H}$, thereby recovering the corresponding Hamiltonian dynamics. Using an integrator, predicted trajectories can be generated from the surrogate $\hat{\mathcal{H}}_{\boldsymbol{\Theta}}$ starting from the initial condition $(\mathbf{p}_{t_0}, \mathbf{q}_{t_0}) \in \Omega$:

$$(\hat{\mathbf{p}}_{t_n}, \hat{\mathbf{q}}_{t_n})_{n=0}^N = \text{Integrator}(\hat{\mathcal{H}}_{\boldsymbol{\Theta}}, (\mathbf{p}_{t_0}, \mathbf{q}_{t_0}), (t_n)_{n=0}^N). \tag{3}$$

Ideally, the surrogate Hamiltonian function should produce predicted trajectories that closely match the observed trajectories given the same initial conditions. To achieve this, we learn $\hat{\mathcal{H}}_{\boldsymbol{\Theta}}$ by minimizing the loss function:

$$\mathcal{L}(\boldsymbol{\Theta}) = \mathbb{E}_{(\mathbf{p}_{t_0}, \mathbf{q}_{t_0}) \sim \mathbb{P}_\Omega} \left[ \sum_{n=0}^N \left[ \|\hat{\mathbf{p}}_{t_n} - \mathbf{p}_{t_n}\|^2 + \|\hat{\mathbf{q}}_{t_n} - \mathbf{q}_{t_n}\|^2 \right] \right], \tag{4}$$

where $(\hat{\mathbf{p}}_{t_n}, \hat{\mathbf{q}}_{t_n})_{n=0}^N$ are generated by $\hat{\mathcal{H}}_{\boldsymbol{\Theta}}$ using Equation (3) and $\mathbb{P}_\Omega$ is some probability distribution over the domain $\Omega \subseteq \mathbb{R}^{2d}$ from which the initial conditions are sampled.

In practise, we have a dataset $\mathcal{D}$ of multiple observed trajectories, where each trajectory $(\mathbf{p}_{t_n}, \mathbf{q}_{t_n})_{n=0}^N$ has an initial condition sampled from a domain $(\mathbf{p}_{t_0}, \mathbf{q}_{t_0}) \sim \mathbb{P}_\Omega$. We can learn a surrogate $\hat{\mathcal{H}}_{\boldsymbol{\Theta}}$ by minimizing the empirical loss function:

$$\hat{\mathcal{L}}(\boldsymbol{\Theta}) = \frac{1}{N|\mathcal{D}|} \sum_{(\mathbf{p}_{t_n}, \mathbf{q}_{t_n})_{n=0}^N \in \mathcal{D}} \left[ \sum_{n=0}^N \left[ \|\hat{\mathbf{p}}_{t_n} - \mathbf{p}_{t_n}\|^2 + \|\hat{\mathbf{q}}_{t_n} - \mathbf{q}_{t_n}\|^2 \right] \right], \tag{5}$$

where $|\mathcal{D}|$ denotes the number of training trajectories and $(\hat{\mathbf{p}}_{t_n}, \hat{\mathbf{q}}_{t_n})$ are generated by $\hat{\mathcal{H}}_{\boldsymbol{\Theta}}$ using Equation (3). By minimizing Equation (5), we learn a surrogate Hamiltonian function $\hat{\mathcal{H}}_{\boldsymbol{\Theta}}$ which can be used to model an unknown Hamiltonian system.

## 3 Methodology

### 3.1 Finite Expression Method for Hamiltonian Systems

H-FEX is a FEX method (Jiang et al., 2023; Liang & Yang, 2022; Song et al., 2024) to build a symbolic approximation using mathematical expression with a finite number of operators, or more simply called finite expressions. A finite expression is a combination of operators (e.g., $\times$, $\exp$, $(\cdot)^2$, $(\cdot)^4$) that form a valid function (e.g., $f(p, q) = \exp(p^2 \times q^4)$). If we limit our operators to unary operators and binary operators, we can represent a finite expression with a binary tree $\mathcal{T}$ consisting of unary and binary nodes. In summary, H-FEX is a method that models the Hamiltonian function using a closed-form expression.

Given a tree $\mathcal{T}$, we can assign operators to the nodes by an operator sequence $\mathbf{e}$, using inorder traversal. In FEX, each element of the operator sequence $\mathbf{e}$ is either from a set of unary operators $\mathbb{U}$ (e.g., $\mathbb{U} = \{\exp, \sin, \text{Id}, (\cdot)^2\}$) or from a set of binary operators $\mathbb{B}$ (e.g., $\mathbb{B} = \{\times, +, \div\}$), where the operators are applied element-wise to vectors. In addition, we parameterize the tree with weights $\boldsymbol{\theta}$, which can be either be scalars of matrices, to expand the class of functions that can be represented by FEX. These weights can be applied between nodes in the tree, influencing intermediate values at various points in the resulting expressions. This enables H-FEX to learn specific constants in the governing equations. In summary, a FEX approximation of the Hamiltonian function $\hat{\mathcal{H}}(\mathbf{p}, \mathbf{q}; \mathcal{T}, \mathbf{e}, \boldsymbol{\theta})$ is a function of $(\mathbf{p}, \mathbf{q})$ with parameters $\mathcal{T}, \mathbf{e}, \boldsymbol{\theta}$.

When H-FEX learns the surrogate Hamiltonian function $\hat{\mathcal{H}}_{\boldsymbol{\Theta}}$ in Equation (5), we let the trainable parameters be $\boldsymbol{\Theta} = (\mathbf{e}, \boldsymbol{\theta})$, where the binary tree structure $\mathcal{T}$ is fixed before training. Minimizing Equation (5) over both discrete and continuous parameters $\boldsymbol{\Theta} = (\mathbf{e}, \boldsymbol{\theta})$ is a difficult task (Köppe, 2012). The operator sequence $\mathbf{e}$ consists of operators selected from some discrete space of $\mathbb{U}$ and $\mathbb{B}$, and the weights $\boldsymbol{\theta}$ are real-valued weights from a continuous space. To solve this mixed optimization problem, we use a search loop, based on reinforcement learning (RL), to identify effective operator sequences $\mathbf{e}$ while optimizing weights $\boldsymbol{\theta}$. The search loop is summarized in the following steps.

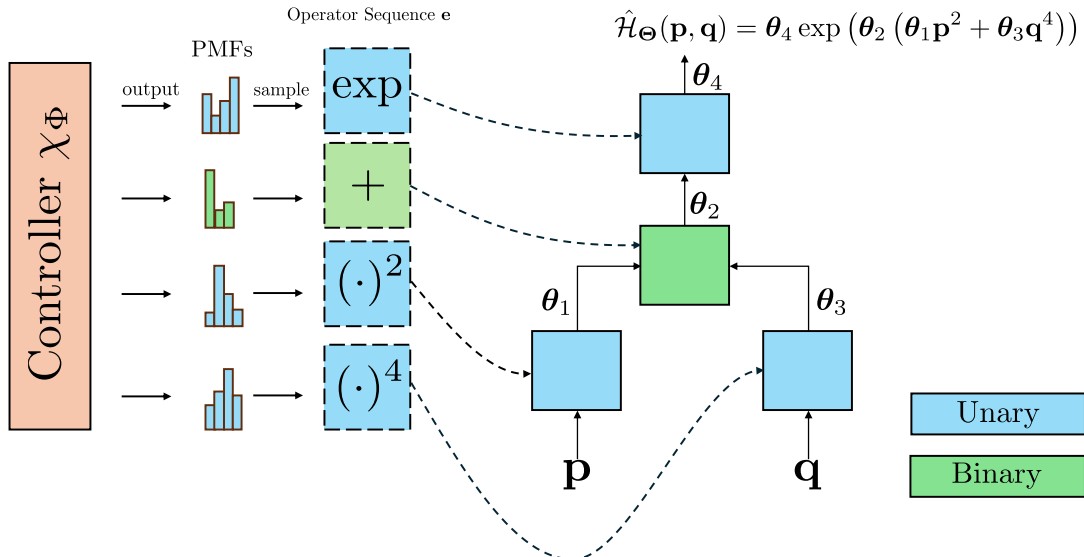

Figure 1: **Generating operator sequences for an H-FEX tree.** To parameterize an H-FEX tree with an operator sequence $\mathbf{e}$, the controller $\chi_\Phi$ generates a PMF for each node. An operator is then sampled from each PMF and assigned to its corresponding node. The resulting H-FEX tree becomes an expression with trainable weights, which are optimized during the scoring process in Equation (6).

1. **Operator sequence generation.** We use RL techniques to identify good candidate operator sequences $\mathbf{e}$. In RL, a controller outputs a policy, a probability mass function (PMF), and an action is sampled according to the probabilities (Kaelbling et al., 1996). In our context, an action corresponds to an operator sequence $\mathbf{e} = (e_1, \ldots, e_Q)$, where $Q$ denotes the number of operator nodes in tree $\mathcal{T}$.

   The controller $\chi_\Phi$ is a fully-connected neural network parameterized by weights $\Phi$ and outputs PMFs $(p_1, \ldots, p_Q)$. For the $i$th node, we sample an operator from the corresponding PMF; i.e., $e_i \sim p_i$, where $1 \leq i \leq Q$, giving us a sampled operator sequence $\mathbf{e}$. As shorthand for the above process, we write $\mathbf{e} \sim \chi_\Phi$ to denote an operator sequence being sampled from the controller. During a single iteration of the search loop, we sample $M$ operator sequences $\mathbf{e}$ from the controller.

   To enhance the exploration of the operator sequence space, we sample using an $\epsilon$-greedy strategy (Dann et al., 2022), where $0 < \epsilon < 1$. With probability $\epsilon$, we sample $e_i$ from a uniform distribution, and with probability $1 - \epsilon$, we sample $e_i$ from PMF $p_i$. A larger $\epsilon$ increases the likelihood of exploring new sequences.

2. **Score computation.** Given an operator sequence $\mathbf{e}$, we compute a score $S(\mathbf{e})$ to quantify the performance of $\mathbf{e}$:

$$S(\mathbf{e}) := \frac{1}{1 + L(\mathbf{e})}, \tag{6}$$

   where $L(\mathbf{e}) := \min_{\boldsymbol{\theta}} \hat{\mathcal{L}}((\mathbf{e}, \boldsymbol{\theta}))$ is the minimum loss optimizing over weights $\boldsymbol{\theta}$ given a fixed operator sequence $\mathbf{e}$, and $\hat{\mathcal{L}}$ is defined in Equation (5).

   As $L(\mathbf{e})$ approaches 0, the score $S(\mathbf{e})$ approaches 1, meaning a larger score indicates a better performing operator sequence $\mathbf{e}$. Defining the score this way prevents excessively large updates to the controller $\chi_\Phi$, keeping the output PMFs stable even when losses $L(\mathbf{e})$ vary wildly across different

operator sequences. In practice, we approximate $L(\mathbf{e})$ with the empirical minimum loss attained when optimizing for a fixed number of steps.

From the previous step, we sampled $M$ operator sequences from the controller: $\mathbf{e}_1, \ldots, \mathbf{e}_M \sim \chi_\Phi$, and compute the corresponding scores $S(\mathbf{e}_1), \ldots, S(\mathbf{e}_M)$ for each sequence individually. During score computation, we typically trade off accuracy for speed when approximating $L(\mathbf{e})$ to expedite the exploration of operator sequences (Song et al., 2024). As a result, the weights $\boldsymbol{\theta}$ learned during this process may not be optimal. We address this issue in the next step.

3. **Candidate Pool.** We maintain a candidate pool to keep track of the highest-scoring operator sequences. The candidate pool stores the parameters $\boldsymbol{\Theta} = (\mathbf{e}, \boldsymbol{\theta})$, obtained after score computation, for the top $K$ scoring operator sequences $\mathbf{e}$ across all iterations of the search loop. Once the search loop concludes, we fine-tune the models in the candidate pool $\mathcal{K}$ by further minimizing $\mathcal{L}((\mathbf{e}, \boldsymbol{\theta}))$ with respect to $\boldsymbol{\theta}$, if necessary.

4. **Controller update.** We use a risk-seeking policy gradient (Petersen et al., 2021) to update the controller $\chi_\Phi$. Standard RL techniques update the weights $\Phi$ of the controller $\chi_\Phi$ by maximizing the objective:

$$\mathcal{J}_s(\Phi) := \mathbb{E}_{\mathbf{e} \sim \chi_\Phi} [S(\mathbf{e})]. \tag{7}$$

This objective function corresponds to maximizing the average score of operator sequences sampled by the controller $\chi_\Phi$. However, in symbolic regression, our interest lies in the best-forming operator sequences rather than the average. Therefore, we instead maximize the expected score of the top $(1 - \nu) \times 100\%$ scores of operator sequences sampled from the controller $\chi_\Phi$:

$$\mathcal{J}(\Phi) := \mathbb{E}_{\mathbf{e} \sim \chi_\Phi} [S(\mathbf{e}) | S(\mathbf{e}) \geq S_\nu(\Phi)], \tag{8}$$

where $S_\nu(\Phi)$ is the $(1 - \nu) \times 100\%$ quantile of the score distribution of operator sequences sampled from $\chi_\Phi$.

We estimate the gradient of Equation (8) using the risk-seeking policy gradient in (Petersen et al., 2021):

$$\nabla_\Phi \mathcal{J}(\Phi) \approx \frac{1}{\nu M} \sum_{j=1}^{M} [S(\mathbf{e}^{(j)}) - \tilde{S}_\nu(\Phi)] \cdot \mathbf{1}\{S(\mathbf{e}^{(j)}) \geq \tilde{S}_\nu(\Phi)\} \cdot \nabla_\Phi \sum_{i=1}^{Q} \log p_i(e_i^{(j)}; \Phi), \tag{9}$$

where $p_i(\cdot; \Phi)$ is the PMF of the $i$th node generated by the controller $\chi_\Phi$, and $\tilde{S}_\nu(\Phi)$ is the empirical $(1 - \nu) \times 100\%$ quantile of the $M$ sampled operator sequences $\mathbf{e}^{(1)}, \ldots, \mathbf{e}^{(M)} \sim \chi_\Phi$. At the end of each iteration of the search loop, we compute the gradient in Equation (9) and update the weights $\Phi$ of the controller $\chi_\Phi$ accordingly.

## 3.2 Interaction Nodes

Hamiltonian systems involving multiple objects (e.g., particles, oscillators) typically have Hamiltonian functions that include interaction terms among position coordinates $\mathbf{q}_i$, and the functional forms of momentum $\mathbf{p}$ and position $\mathbf{q}$ often differ. For example, in the N-body problem (Aarseth, 2003), the Hamiltonian function is given by $\mathcal{H}(\mathbf{p}_1, \ldots, \mathbf{p}_N, \mathbf{q}_1, \ldots, \mathbf{q}_N) = \sum_{i=1}^{N} c_0 \|\mathbf{p}_i\|^2 - \sum_{i<j} \frac{c_1}{\|\mathbf{q}_i - \mathbf{q}_j\|}$, and in coupled harmonic oscillators (Makarov, 2018), it is $\mathcal{H}(\mathbf{p}_1, \mathbf{p}_2, \mathbf{q}_1, \mathbf{q}_2) = c_0 \mathbf{p}_1^2 + c_1 \mathbf{p}_2^2 + c_2 \mathbf{q}_1^2 + c_3 \mathbf{q}_2^2 + c_4 \mathbf{q}_1 \mathbf{q}_2$, where $c_i$ are constants. Many Hamiltonian functions exhibit these structures, so we design the H-FEX tree to treat $\mathbf{p}$ and $\mathbf{q}$ separately and introduce a dedicated interaction node to effectively represent interaction terms.

The interaction node enables an H-FEX tree to represent pairwise interaction terms. Similar to unary and binary nodes, the interaction node is assigned an operator sampled from a set $\mathbb{I}$. Each operator in $\mathbb{I}$ maps a pairwise combination of elements to a scalar value, with exception of a "no interaction" operator (Id) that is functionally equivalent to a unary operator with the identity operator Id. If the interaction node takes a vector input $\mathbf{x} \in \mathbb{R}^d$, then the set $\mathbb{I}$ may include operators such as $(x_i - x_j)$, $(x_i - x_j)^2$, $x_i x_j$, where $x_i$

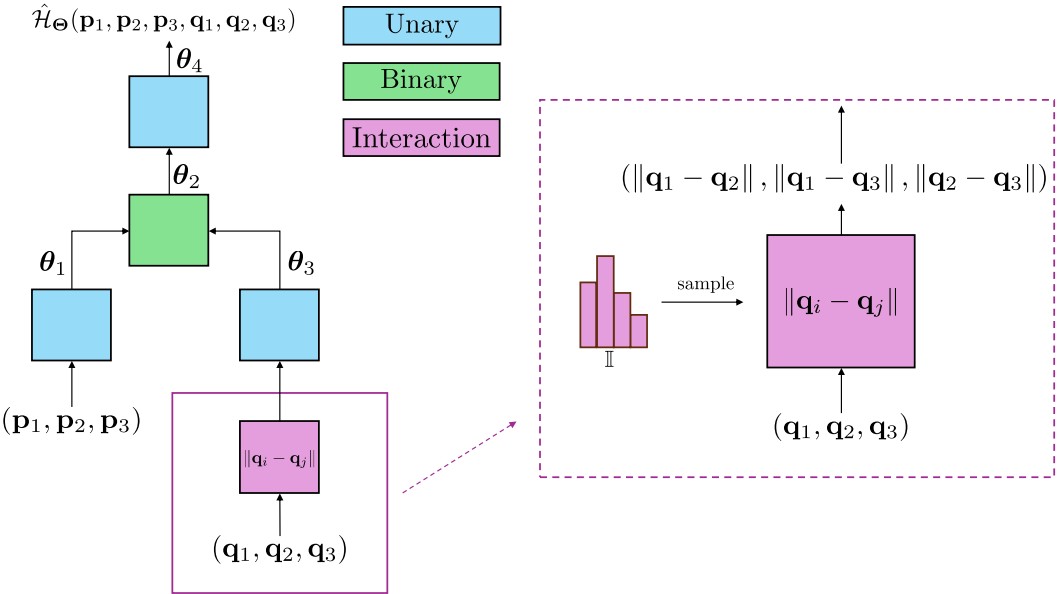

Figure 2: **H-FEX tree with an interaction node.** This example H-FEX tree includes an interaction node that operates on the position coordinates $(\mathbf{q}_1, \mathbf{q}_2, \mathbf{q}_3)$. The interaction node is parameterized by an operator sampled from a PMF over the set of interaction operators $\mathbb{I}$. The set $\mathbb{I}$ contains operators that maps pairwise combinations of elements to a scalar value. In this instance, the sampled interaction operator is $\|\mathbf{q}_i - \mathbf{q}_j\|$ for $i \neq j$, and the node outputs all pairwise interactions between position coordinates.

and $x_j$ are distinct elements of $\mathbf{x}$ (i.e., $i \neq j$). More generally, if input is a matrix $\mathbf{X} \in \mathbb{R}^{r \times d}$, then the set $\mathbb{I}$ may include operators such as $\|\mathbf{x}_i - \mathbf{x}_j\|$, $\|\mathbf{x}_i - \mathbf{x}_j\|^2$, $\|\mathbf{x}_i \odot \mathbf{x}_j\|$, where $\|\cdot\|$ denotes the Euclidean norm and $\odot$ denotes elementwise multiplication. These interaction operators map each pairwise combination of the $r$ column vectors in $X$ to a scalar. In essence, the interaction node enables the H-FEX tree to model interaction terms, allowing it to capture complex Hamiltonian functions that involve pairwise interactions. Figure 2 shows an example of an H-FEX tree with an interaction node to represent interaction terms among the position coordinates.

### 3.3 Training and evaluation for H-FEX

Integrators enable the simulation of trajectories of candidate H-FEX trees during both training and evaluation. Once an operator sequence $\mathbf{e}$ parameterizes an H-FEX tree, the resulting expression yields a Hamiltonian surrogate $\mathcal{H}_{\Theta}$, which can then be integrated into the predicted trajectories generated. During training, these trajectories are compared to observed data to calculate the empirical loss in Equation (5), which is also used to calculate a score of the operator sequence $\mathbf{e}$ in Equation (6). During the evaluation, a trained Hamiltonian surrogate $\mathcal{H}_{\Theta}$ is used to simulate the trajectories of a given initial condition. At a high level, the prediction of the trajectory is summarized in Equation (3), but in this section, we provide a more detailed explanation in the following.

To reduce numerical error during integration, we adopt a multi-step integration scheme similar to that used in (Chen et al., 2019). Rather than integrating from $t_{n-1}$ to $t_n$ in a single step, each interval $[t_{n-1}, t_n]$ is subdivided into uniform $K$ substeps $(t_{n_k})_{k=0}^K$. During evaluation, trajectories are recursively generated by integrating forward from the previously predicted state:

$$(\hat{\mathbf{p}}_{t_{n+1}}, \hat{\mathbf{q}}_{t_{n+1}}) = \text{Integrator}(\hat{\mathcal{H}}_{\Theta}, (\hat{\mathbf{p}}_{t_n}, \hat{\mathbf{q}}_{t_n}), (t_{n_k})_{k=0}^K) \quad \text{for } n = 0, 1, \ldots, N-1, \tag{10}$$

where $(\hat{\mathbf{p}}_{t_0}, \hat{\mathbf{q}}_{t_0})$ is set to the initial condition. In contrast, training uses the previous state from the observed trajectory at each step:

$$(\hat{\mathbf{p}}_{t_{n+1}}, \hat{\mathbf{q}}_{t_{n+1}}) = \text{Integrator}(\hat{\mathcal{H}}_{\boldsymbol{\Theta}}, (\mathbf{p}_{t_n}, \mathbf{q}_{t_n}), (t_{n_k})_{k=0}^{K}) \quad \text{for } n = 0, 1, \ldots, N-1, \tag{11}$$

which prevents the propogation of integration errors during optimization, leading to more stable training. In Equation (10) and Equation (11), we use the Leapfrog integrator in Algorithm 1 for separable systems and the second-order Runge-Kutta (RK2) method in Algorithm 2 for non-separable systems. Additional details on integrators are provided in Appendix A.

## 4 Numerical results

In this section, we evaluate the performance of H-FEX by comparing it with existing methods on two problems: a nonseparable Hamiltonian system (Wu et al., 2020) and the three-body problem (Gowers, 2008). In both cases, the objective is to learn the underlying Hamiltonian function directly from trajectory data, thereby recovering the corresponding Hamiltonian system.

The experiments are conducted as follows. We begin by generating trajectores from the true system and use them as training data for each method. After training, the learned models are evaluated on a separate test set consisting of trajectories not seen during training. Using the initial states of the test data, we simulate predicted trajectories from the learned surrogate of each method and compare them to the corresponding true test trajectories. The test trajectories span significantly longer time intervals than those used for training, allowing us to assess each model's ability to generalize and remain stable over extended predictions. We evaluate the methods based on the accuracy of the predicted trajectories over time and the degree to which they conserve energy.

We assess the accuracy of a predicted trajectory $(\hat{\mathbf{p}}_{t_i}, \hat{\mathbf{q}}_{t_i})_{i=0}^{N_{\text{test}}}$ by calculating the mean squared error (MSE) at each time step:

$$\text{MSE}(t) := \frac{1}{2d} \left( ||\hat{\mathbf{p}}_t - \mathbf{p}_t||^2 + ||\hat{\mathbf{q}}_t - \mathbf{q}_t||^2 \right), \tag{12}$$

where $\mathbf{p}_t, \hat{\mathbf{p}}_t, \mathbf{q}_t, \hat{\mathbf{q}}_t \in \mathbb{R}^d$. For each test trajectory, we compute $\text{MSE}(t)$ at each time point to analyze how the prediction error evolves over time. Additionally, we assess the ability of each method to conserve energy by computing the relative error, denoted by $E_{\text{rel}}(t)$ in the energy over time:

$$E_{\text{rel}}(t) := \frac{|\mathcal{H}(\hat{\mathbf{p}}_t, \hat{\mathbf{q}}_t) - \mathcal{H}(\hat{\mathbf{p}}_{t_0}, \hat{\mathbf{q}}_{t_0})|}{|\mathcal{H}(\hat{\mathbf{p}}_{t_0}, \hat{\mathbf{q}}_{t_0})|}, \tag{13}$$

where $|\cdot|$ denotes absolute value and $\mathcal{H}$ is the true Hamiltonian function and $(\hat{\mathbf{p}}_{t_0}, \hat{\mathbf{q}}_{t_0})$ is the initial state. If the predictions $(\hat{\mathbf{p}}_t, \hat{\mathbf{q}}_t)$ conserve energy well, the Hamiltonian $\mathcal{H}(\hat{\mathbf{p}}_t, \hat{\mathbf{q}}_t)$ should remain approximately constant over time. Consequently, the relative error in Equation (13) should remain close to zero.

### 4.1 Non-Separable Hamiltonian System

We compare H-FEX with SINDy (Brunton et al., 2016) on a non-separable system from (Wu et al., 2020). The Hamiltonian system is characterized by the non-separable Hamiltonian function $\mathcal{H} : \mathbb{R}^2 \to \mathbb{R}$ defined by:

$$\mathcal{H}(p, q) = \exp(-\alpha_1 p^2 - \alpha_2 q^4), \tag{14}$$

where $\alpha_1 = 1$ and $\alpha_2 = 1.1$. By Equation (1), this Hamiltonian function yields the following dynamical system:

$$\begin{aligned} \frac{dp}{dt} &= 4\alpha_2 q^3 \exp(-\alpha_1 p^2 - \alpha_2 q^4), \\ \frac{dq}{dt} &= -2\alpha_1 p \exp(-\alpha_1 p^2 - \alpha_2 q^4). \end{aligned} \tag{15}$$

For the data, the initial states $(p_{t_0}, q_{t_0})$ are uniformly sampled from the domain $\Omega = [-1, 1]^2$. For the training data, we discretize the time interval $[0, 3]$ using 30 points with a uniform timestep of 0.1. Using the adaptive step integrator RK45 (Fehlberg, 1969), we generate 120 trajectories on the interval $[0, 3]$ from different initial states using the true dynamical system Equation (15). For the testing data, we use the same data generating process, but instead use a longer time interval of $[0, 60]$, discretized with the same timestep of 0.1.

We state all the details regarding the structure and operator search loop of H-FEX for this non-separable system. The H-FEX binary tree consists of three unary nodes and one binary node, with 4 weights to scale the output of each node (see Figure 7). We use the following unary and binary operator dictionaries:

$$\mathbb{U} = \{\text{Id}, (\cdot)^2, (\cdot)^3, (\cdot)^4, \exp, \sin, (\cdot)^{-1}\}, \qquad \mathbb{B} = \{+, \times, -, \div\},$$

where "Id" denotes the identity operator. We run the search loop for 100 iterations, and each iteration, we generate 15 FEX trees. For operator sequence generation, we use an $\epsilon$-greedy strategy (Dann et al., 2022) with $\epsilon = 0.2$ to sample 15 operator sequences from the PMFs given by the controller. During score computation, we minimize the loss function Equation (5), for each FEX tree, using the Adam optimizer with its default moment parameters $\beta_1 = 0.9, \beta_2 = 0.999$ (Kingma & Ba, 2014) and a learning rate of 0.1 for 150 steps. The default values of $\beta_1$ and $\beta_2$ are used in all our experiments. Since the system is non-separable, we generate trajectories during training with Equation (11) using RK2 (Butcher, 1987), a traditional integrator, with $K = 20$ substeps. The controller is a small fully-connected neural network deterministically mapping the zero vector to PMFs for each node, and the weights of the controller are updated using the risk-seeking policy gradient (Petersen et al., 2021) with $\nu = 0.25$. During the search loop, we maintain a candidate pool to save the 15 top-scoring FEX models, and after the search loop, we fine-tune each of the models using Adam with a learning rate of 0.001 for 300 steps. After the fine-tuning, we select the H-FEX tree with the highest score to compare with SINDy.

The resulting H-FEX tree has an operator sequence $\mathbf{e} = ((\cdot)^2, +, (\cdot)^4, \exp)$ (inorder traversal) and weights $\boldsymbol{\theta} = (0.9236, 1.0159, -1.0830, 1.0000)$, yielding the closed form approximation of the Hamiltonian function:

$$\hat{\mathcal{H}}_{\boldsymbol{\Theta}}(p, q) = 1.0000 \exp(-1.0830(0.9236p^2 + 1.0159q^4))$$
$$= \exp(-1.0003p^2 - 1.1002q^4).$$

Comparing the approximation with Equation (14), H-FEX correctly identifies the true operators and learns weights that simplify to values close to the true values of $\alpha_1$ and $\alpha_2$. Using Equation (3) on the 30 test trajectories, we generate predicted trajectories on the interval $[0, 60]$ given the initial states of the testing set. We then use these predicted trajectories to compute the MSE and relative energy change over time and compare with SINDy in Figure 3. H-FEX maintains low trajectory error even at time points far beyond the training interval, whereas SINDY's trajectory error steadily increases over time. Additionally, H-FEX preserves the initial energy throughout all time points, while SINDy quickly deviates from the initial energy.

On the other hand, SINDy yields a closed-form expression for $(\frac{dp}{dt}, \frac{dq}{dt})$ directly:

$$\frac{dp}{dt} = 0.1517q - 0.6835p^2q + 4.0850q^3 + 0.2606p^4q - 0.8336p^2q^3 - 2.7519q^5,$$
$$\frac{dq}{dt} = -1.9010p + 1.5567p^3 + 0.7090pq^2 - 0.3652p^5 - 0.5934p^3q^2 + 0.4568pq^4.$$

This learned expression differs significantly from the true dynamical system in Equation (15). The source of this discrepancy becomes clear upon brief examination of the SINDy algorithm (Brunton et al., 2016). SINDy constructs expressions by using a predefined library of candidate functions (e.g., polynomials, trigonometric terms) and selecting a sparse linear combination that best fits the observed data $(\mathbf{p}_{t_n}, \mathbf{q}_{t_n})_{n=0}^{N}$. In this experiment, the library includes polynomial terms up to degree 6. Because SINDy can only produce linear combinations of these candidate functions, it struggles to capture complex nonlinear structures such as those present in Equation (15).

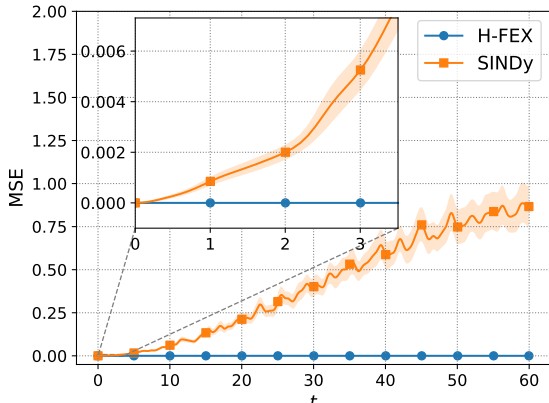 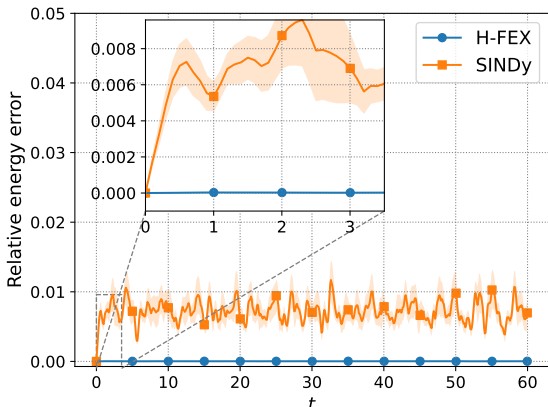

Figure 3: **Non-separable results.** FEX and SINDy are trained using trajectories on $[0, 3]$ and evaluated using 30 test trajectories on $[0, 60]$. **Left**: MSE over time, defined in Equation (12). **Right**: Relative energy change over time, defined in Equation (13). FEX exhibits virtually no MSE and energy drift over longer time horizons, in stark contrast to SINDy, which accumulates error over time and deviates from the initial energy almost immediately.

## 4.2 Three-Body Problem

The three-body problem (Gowers, 2008) is a classical problem in physics and celestial mechanics that involves predicting the motion of three bodies interacting under their mutual gravitational forces. In our experiments, we consider three point masses interacting in two-dimensional space. For $1 \leq i \leq 3$, the $i$th body has its momentum denoted by $\mathbf{p}_i \in \mathbb{R}^2$ and position denoted by $\mathbf{q}_i \in \mathbb{R}^2$, so the system can be represented by coordinates $(\mathbf{p}_1, \mathbf{p}_2, \mathbf{p}_3, \mathbf{q}_1, \mathbf{q}_2, \mathbf{q}_3) \in \mathbb{R}^{12}$. The corresponding Hamiltonian function is $\mathcal{H} : \mathbb{R}^{12} \to \mathbb{R}$ defined by:

$$\mathcal{H}(\mathbf{p}_1, \mathbf{p}_2, \mathbf{p}_3, \mathbf{q}_1, \mathbf{q}_2, \mathbf{q}_3) = \frac{\|\mathbf{p}_1\|^2}{2m_1} + \frac{\|\mathbf{p}_2\|^2}{2m_2} + \frac{\|\mathbf{p}_3\|^2}{2m_3} - \frac{Gm_1m_2}{\|\mathbf{q}_1 - \mathbf{q}_2\|} - \frac{Gm_1m_3}{\|\mathbf{q}_1 - \mathbf{q}_3\|} - \frac{Gm_2m_3}{\|\mathbf{q}_2 - \mathbf{q}_3\|}, \qquad (16)$$

where $\|\cdot\|$ denotes the Euclidean norm, $m_i \in \mathbb{R}$ denotes the mass of the $i$th body, and $G \in \mathbb{R}$ denotes the gravitational constant. In our experiments, we set $m_i = 1$ for each $i$ and $G = 1$. To generate initial states, we place the three point masses on a randomly chosen circle (with radius uniformly drawn from $[0.9, 1.2]$) by selecting a random point for the first point mass and then rotating it by $120°$ for the other two. Each point mass is given a momumtum perpendicular to its position vector to produce a perfect circular orbit. Finally, we perturb these momenta by multiplying them by a random factor uniformly drawn from $[0.8, 1.2]$. For testing, we generate 30 trajectories on the time interval $[0, 30]$ with a timestep of 0.1 using the integrator RK45 (Fehlberg, 1969) until a relative error less than $10^{-9}$ is attained.

We consider two different training datasets: the first with trajectories on a smaller time interval $[0, 3]$, and the second with trajectories on a larger time interval $[0, 7]$, each with the same timestep of 0.1. As the trajectories are being generated, collisions between two or more point masses will inevitibly occur, with more potential for collisions on larger time intervals. Therefore, trajectories on the interval $[0, 3]$ tend to have less collisions than trajectories on the interval $[0, 7]$. When collisions occur, two or more position vectors become very close, leading to a singularity in Equation (16). In other words, collisions create stiff regions in a trajectory, where integration becomes more challenging. To quantify the stiffness of the two training datasets, we calculate the stiffness-aware index (SAI) from (Liang et al., 2021; Huang et al., 2023) defined by:

$$\mathrm{SAI}(\mathbf{p}_{t_i}, \mathbf{q}_{t_i}) := \frac{1}{\|(\mathbf{p}_{t_i}, \mathbf{q}_{t_i})^\top\|} \cdot \frac{\|(\mathbf{p}_{t_{i+1}}, \mathbf{q}_{t_{i+1}})^\top - (\mathbf{p}_{t_i}, \mathbf{q}_{t_i})^\top\|}{t_{i+1} - t_i}, \qquad (17)$$

where $\|\cdot\|$ is the Euclidean norm and $(\mathbf{p}_{t_i}, \mathbf{q}_{t_i})^\top \in \mathbb{R}^{2d}$ is the state of the trajectory at time $t_i$. Higher SAI values correspond to trajectory segments that exhibit greater stiffness. In Figure 4, we compute the densities

of SAI for both datasets and then plot on log scale to emphasize the differences. The dataset on $[0, 7]$ has regions with larger SAI compared to the dataset on $[0, 3]$, supporting our intuition that trajectories on the interval $[0, 7]$ are more challenging to integrate due to increased presence of collisions. For both training datasets, we use the same H-FEX structure and training hyperparameters.

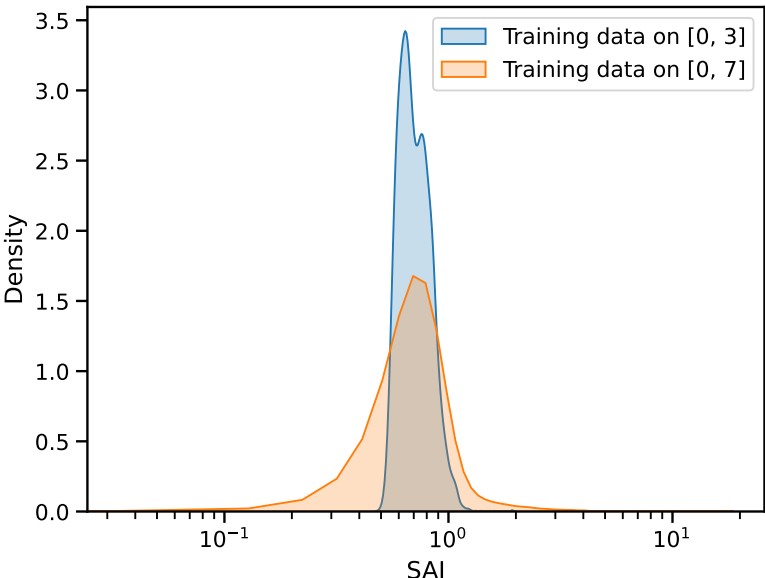

Figure 4: **SAI Density Plot.** The stiffness-aware index (SAI), defined in Equation (17), is used to empirically quantify the stiffness of training trajectories under two different scenarios in the three-body problem. We compute densities of SAI and then plot on log scale for two datasets: one containing trajectories from $[0, 3]$, the second from $[0, 7]$. Higher SAI values correspond to stiffer regions of the trajectory. As shown in the density plots, trajectories on $[0, 7]$ exhibit stiff regions not present in $[0, 3]$, primarily due to collisions occurring over the longer interval.

The structure of an H-FEX tree consists of three unary nodes, one binary node, and one interaction node to produce interaction terms among the coordinate terms $(\mathbf{q}_1, \mathbf{q}_2, \mathbf{q}_3)$, with weights after each binary and unary node (see Figure 7). For the unary, binary, and interaction nodes, we use the following operator dictionaries:

$$\mathbb{U} = \{\mathrm{Id}, (\cdot)^2, (\cdot)^3, \exp, \sin, (\cdot)^{-1}\},$$
$$\mathbb{B} = \{+, \times, -\},$$
$$\mathbb{I} = \{\|\mathbf{q}_i - \mathbf{q}_j\|^2, \|\mathbf{q}_i - \mathbf{q}_j\|, \|\mathbf{q}_i \odot \mathbf{q}_j\|^2, \|\mathbf{q}_i \odot \mathbf{q}_j\|\} \quad \text{for } i \neq j.$$

We run the search loop for 100 iterations, and for each iteration, we sample 10 operator sequences from the controller using an $\epsilon$-strategy (Dann et al., 2022) with $\epsilon = 0.2$. We compute the score by minimizing Equation (5) using Adam (Kingma & Ba, 2014) with a learning rate of 0.1 for 150 steps. Predicted trajectories are generated during training using Equation (11) with $K = 20$ substeps using leapfrog (Birdsall & Langdon, 2018), a symplectic integrator. The weights of the controller are updated using the risk-seeking policy gradient in Equation (9) with $\nu = 0.25$. The candidate pool saves the top 15 highest-scoring H-FEX trees, and after the search loop, we fine-tune each of the trees using Adam with a learning rate of 0.001 for 300 steps. After fine-tuning, we select the highest scoring H-FEX tree to compare with other models.

We show only the results of H-FEX when trained with trajectories on $[0, 7]$, as the results of H-FEX when trained with trajectories on $[0, 3]$ are similar for H-FEX. The highest-scoring H-FEX tree has an operator

sequence of $\mathbf{e} = \left((\cdot)^2, +, \|\mathbf{q}_i - \mathbf{q}_j\|, (\cdot)^{-1}, \mathrm{Id}\right)$, where $(\cdot)^2, (\cdot)^{-1}, \mathrm{Id} \in \mathbb{U}$, $+ \in \mathbb{B}$, and $\|\mathbf{q}_i - \mathbf{q}_j\| \in \mathbb{I}$. This yields the closed form approximation of the Hamiltonian function:

$$
\begin{aligned}
\hat{\mathcal{H}}_{\boldsymbol{\Theta}}(\mathbf{p}_1, \mathbf{p}_2, \mathbf{p}_3, \mathbf{q}_1, \mathbf{q}_2, \mathbf{q}_3) = {} & \left(0.5005 p_{1,x}^2 + 0.4981 p_{1,y}^2 + 0.4996 p_{2,x}^2 + 0.4985 p_{2,y}^2 + 0.4976 p_{3,x}^2 + 0.5024 p_{3,y}^2\right) \\
& + \left(\frac{-0.9985}{\|\mathbf{q}_1 - \mathbf{q}_2\|} + \frac{-0.9970}{\|\mathbf{q}_1 - \mathbf{q}_3\|} + \frac{-0.9949}{\|\mathbf{q}_2 - \mathbf{q}_3\|}\right),
\end{aligned}
$$

where for each body, $\mathbf{p}_i = (p_{i,x}, p_{i,y})^\top$ is the momentum on the xy plane. The equation learned by H-FEX closely matches Equation (16), as we let $G = 1$ and $m_i = 1$ for each body in our experiments. With Equation (3), we use 30 intial conditions from the test dataset to generate predicted trajectories on the interval $[0, 30]$ using the highest-scoring H-FEX tree and compare to other models for both training scenarios.

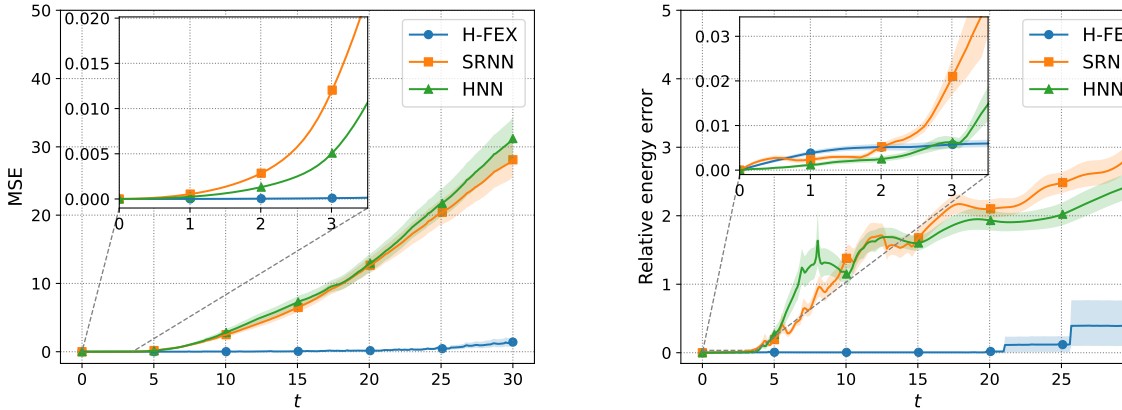

Figure 5: **Three-body problem results with training on** $[0, 3]$**.** H-FEX, SRNN, and HNN are trained using trajectories on $[0, 3]$ and evaluated using 30 test trajectories on $[0, 30]$. **Left**: MSE over time, defined in Equation (12). **Right**: Relative energy change over time, defined in Equation (13). H-FEX shows slower growth in both MSE and enery drift over time compared to SRNN and HNN.

In Figure 5, we plot the MSE and relative energy change over time for models trained with trajectories on the interval $[0, 3]$. We compare H-FEX with SRNN (Chen et al., 2019) and HNN (Greydanus et al., 2019), which are both methods that use a neural network as a surrogate for the Hamiltonian function so an interpretable closed-form expression is not attainable. H-FEX maintains accurate trajectories even at large time points, and it preserves the initial energy for long periods of time with deviations in the energy only occuring at later time points.

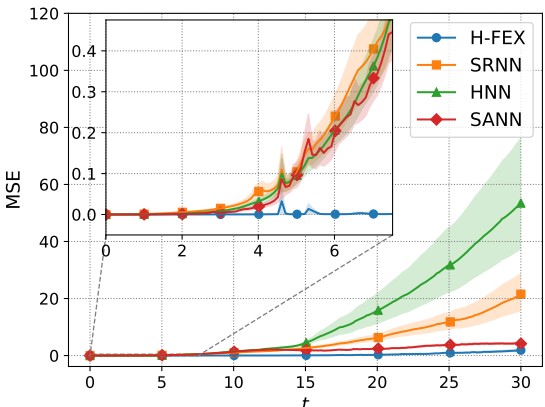 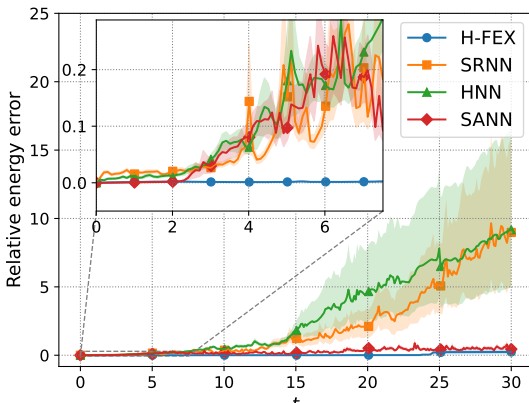

Figure 6: **Three-body problem results with training on [0, 7].** H-FEX, SRNN, HNN, and SANN are trained using trajectories on $[0, 7]$, which are more stiff due to collisions, and evaluated using 30 test trajectories on $[0, 30]$. **Left**: MSE over time, defined in Equation (12). **Right**: Relative energy change over time, defined in Equation (13). Despite the increased difficult of the training data, H-FEX maintains the lowest MSE and energy drift among all methods.

In Figure 6, we plot the MSE and relative energy change over time for models trained with trajectories on the interval $[0, 7]$. Here, we compare H-FEX with SRNN, HNN, and SANN (Liang et al., 2021), where SANN is a neural network surrogate for the Hamiltonian function that can more accurately learn stiff regions of the training trajectories. Since the dataset on $[0, 7]$ contains more collisions, the MSE and the relative energy change of HNN and SRNN diverge faster than when trained using the dataset on $[0, 3]$. H-FEX still produces the most accurate trajectories and preserves energy the best, while providing a closed-form approximation of the Hamiltonian function.

## 5    Discussion and Limitations

In this work, we only consider H-FEX with fixed tree structures, making its performance dependent on the choice of tree structure. If the tree structure cannot represent the form of the true expression, then H-FEX will not be able to recover it. However, by incorporating identity operators into the unary and interaction nodes, as show in fig. 7 (b) and fig. 9, larger trees can produce expressions functionally equivalent to those of smaller trees. Consequently, the tree structure does not need to exactly match the true expression; it only needs to contain a subtree that contains the true expression. While larger trees increase expressivity, they also enlarge the operator search space, requiring more iterations of the search loop to realiably identify good operator sequences. Parallelizing tree scoring can help mitigate this cost, but only up to moderate tree sizes, as the operator search space would grow faster.

## 6    Conclusion

This paper introduces H-FEX, an adaptation of FEX for Hamiltonian systems. H-FEX represents the Hamiltonian function as a tree of operators, enabling it to learn complex mathematical expressions. We modify the FEX search loop for Hamiltonian systems and introduce interaction nodes to better capture interaction terms common in coupled systems. Numerical experiments show that H-FEX accurately identifies the operators and weights needed to construct a tree that closely approximates the true Hamiltonian function. Moreover, the predicted trajectories are highly accurate and preserve energy over long time horizons, demonstrating H-FEX's capability in accurately modeling complex Hamiltonian systems.

Like symbolic regression methods such as SINDy, H-FEX depends on the choice of operators in its dictionaries, which significantly impacts the accuracy of the learned closed-form solution. If essential operators are

missing, H-FEX's accuracy will suffer. Additionally, H-FEX requires a predefined tree structure with operator nodes and weights, shaping the resulting equations. When prior knowledge of the Hamiltonian system is available, it can guide the selection of operators and tree structure to improve modeling. However, without prior knowledge, one must experiment with various operators and structures. In future work, we believe it is fruitful to develop adaptive methods that dynamically adjust node and weight placement, allowing H-FEX to learn complex physics without a predefined tree structure.

## Acknowledgement

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

# A   Additional Integrator Details

This section provides further information on the integrators used during both evaluation (see Equation (10)) and training (see Equation (11)). To reduce notational clutter, we use the shorthand $(\mathbf{p}_n, \mathbf{q}_n)$ to denote $(\mathbf{p}_{t_n}, \mathbf{q}_{t_n})$. The notation $\hat{\mathcal{H}}_p(\mathbf{p}_k, \mathbf{q}_k)$ denotes the derivative of the surrogate Hamiltonian $\hat{\mathcal{H}}$ with respect to $\mathbf{p}$, evaluated at the point $(\mathbf{p}_k, \mathbf{q}_k)$. This derivative can be computed exactly and quickly using automatic differentiation.

The multi-step integration algorithms for advancing a single time step using the Leapfrog and RK2 integrators are shown in Algorithm 1 and Algorithm 2 respectively. These algorithms take the surrogate of the Hamiltonian function $\hat{\mathcal{H}}$ and the current state $(\tilde{\mathbf{p}}_n, \tilde{\mathbf{q}}_n)$, integrates for $K$ substeps, and outputs the predicted next state $(\hat{\mathbf{p}}_{n+1}, \hat{\mathbf{q}}_{n+1})$. During evaluation, the current state $(\tilde{\mathbf{p}}_n, \tilde{\mathbf{q}}_n)$ is set to the previous predicted state $(\hat{\mathbf{p}}_n, \hat{\mathbf{q}}_n)$, while during training, the current state $(\tilde{\mathbf{p}}_n, \tilde{\mathbf{q}}_n)$ is set to the previous observed state $(\mathbf{p}_n, \mathbf{q}_n)$. It should be noted that the integration for each time step is parallelizable during training, greatly reducing the runtime for computing a score of an H-FEX tree.

The Leapfrog and RK2 integrators have different properties and assumptions. The Leapfrog integrator belongs to the class of symplectic integrators, which simulate trajectories with less energy drift, but assumes the Hamiltonian function is separable, i.e., $\mathcal{H}(\mathbf{p}, \mathbf{q}) = K(\mathbf{p}) + U(\mathbf{q})$. On the other hand, the RK2 integrator does not rely on the separability of the Hamiltonian but lacks the energy drift reduction properties of Leapfrog integration. Consequently, we first attempt to use Leapfrog during training, but if many H-FEX trees have trouble optimizing, we switch to RK2 as a fallback.

---

**Algorithm 1** Multi-step Leapfrog integrator for Hamiltonian systems

---

    **function** LEAPFROGINTEGRATOR($\hat{\mathcal{H}}$, $(\tilde{\mathbf{p}}_n, \tilde{\mathbf{q}}_n)$, $(t_{n_k})_{k=1}^{K}$)

        $\Delta t \leftarrow t_{n_2} - t_{n_1}$                                      ▷ $(t_{n_k})_{k=1}^{K}$ are uniform substeps

        $(\mathbf{p}_0, \mathbf{q}_0) \leftarrow (\tilde{\mathbf{p}}_n, \tilde{\mathbf{q}}_n)$

        **for** $k = 0$ to $K - 1$ **do**

            $\mathbf{p}_{k+\frac{1}{2}} \leftarrow \mathbf{p}_k - \frac{\Delta t}{2}\hat{\mathcal{H}}_q(\mathbf{p}_k, \mathbf{q}_k)$            ▷ Leapfrog update adapted for Hamiltonian systems

            $\mathbf{q}_{k+1} \leftarrow \mathbf{q}_k + \Delta t\hat{\mathcal{H}}_p(\mathbf{p}_{k+\frac{1}{2}}, \mathbf{q}_k)$

            $\mathbf{p}_{k+1} \leftarrow \mathbf{p}_{k+\frac{1}{2}} - \frac{\Delta t}{2}\hat{\mathcal{H}}_q(\mathbf{p}_{k+\frac{1}{2}}, \mathbf{q}_{k+1})$

        $(\hat{\mathbf{p}}_{n+1}, \hat{\mathbf{q}}_{n+1}) \leftarrow (\mathbf{p}_K, \mathbf{q}_K)$

        **return** $(\hat{\mathbf{p}}_{n+1}, \hat{\mathbf{q}}_{n+1})$

---

---

**Algorithm 2** Multi-step RK2 integrator for Hamiltonian systems

---

    **function** RK2INTEGRATOR($\hat{\mathcal{H}}$, $(\tilde{\mathbf{p}}_n, \tilde{\mathbf{q}}_n)$, $(t_{n_k})_{k=1}^{K}$)

        $\Delta t \leftarrow t_{n_2} - t_{n_1}$                                        ▷ $(t_{n_k})_{k=1}^{K}$ are uniform substeps

        $(\mathbf{p}_0, \mathbf{q}_0) \leftarrow (\tilde{\mathbf{p}}_n, \tilde{\mathbf{q}}_n)$

        **for** $k = 0$ to $K - 1$ **do**

            $r_1^{(p)} \leftarrow -\hat{\mathcal{H}}_q(\mathbf{p}_k, \mathbf{q}_k)$                ▷ RK2 update adapted for Hamiltonian systems

            $r_1^{(q)} \leftarrow \hat{\mathcal{H}}_p(\mathbf{p}_k, \mathbf{q}_k)$

            $r_2^{(p)} \leftarrow -\hat{\mathcal{H}}_q(\mathbf{p}_k + \frac{\Delta t}{2}r_1^{(p)}, \mathbf{q}_k + \frac{\Delta t}{2}r_1^{(q)})$

            $r_2^{(q)} \leftarrow \hat{\mathcal{H}}_p(\mathbf{p}_k + \frac{\Delta t}{2}r_1^{(p)}, \mathbf{q}_k + \frac{\Delta t}{2}r_1^{(q)})$

            $\mathbf{p}_{k+1} \leftarrow \mathbf{p}_k + \Delta t\, r_2^{(p)}$

            $\mathbf{q}_{k+1} \leftarrow \mathbf{q}_k + \Delta t\, r_2^{(q)}$

        $(\hat{\mathbf{p}}_{n+1}, \hat{\mathbf{q}}_{n+1}) \leftarrow (\mathbf{p}_K, \mathbf{q}_K)$

        **return** $(\hat{\mathbf{p}}_{n+1}, \hat{\mathbf{q}}_{n+1})$

---

## B  H-FEX Search Loop

In Algorithm 3, we show a detailed pseudocode of the H-FEX search loop. The search loop seeks the optimal operator sequence in $E$ iterations. Each iteration, PMFs are generated by the controller $\chi_\Phi$, and $B$ H-FEX trees are constructed from sampled operator sequences. After constructing an H-FEX tree, the weights are optimized for $L$ iterations. The score of each of the $B$ operator sequences can then be used to compute the risk seeking policy gradient in Equation (9).

---

**Algorithm 3** H-FEX search loop high-level pseudocode

---

**procedure** SEARCHLOOP($\mathcal{T}$)                                                                  ▷ Assume a fixed tree $\mathcal{T}$
  **for** 1 to $E$ **do**
    PMFs $\leftarrow \chi_\Phi(\mathbf{0})$
    **for** 1 to $B$ **do**                          ▷ Constructing multiple trees with sampled operator sequences
      $\mathbf{e} \sim$ PMFs
      $\boldsymbol{\theta} \sim$ UniformDistribution                                      ▷ Initializing network weights
      Construct H-FEX tree $\hat{\mathcal{H}}(\cdot, \cdot; \mathcal{T}, \mathbf{e}, \boldsymbol{\theta})$
      **for** 1 to $L$ **do**                                                    ▷ Optimizing weights
        $(\hat{\mathbf{p}}, \hat{\mathbf{q}}) \leftarrow$ Integrator($\hat{\mathcal{H}}, (\mathbf{p}_0, \mathbf{q}_0), )$
        Compute loss $\mathcal{L}(\boldsymbol{\Theta})$
        Update tree weights $\boldsymbol{\theta}$
      Compute score $S(\mathbf{e})$
      **if** $S(\mathbf{e})$ among top $K$ scoring models in the pool **then**
        Add $\hat{\mathcal{H}}$ to the pool
    Compute risk-seeking policy gradient $\nabla_\Phi \mathcal{J}(\Phi)$
    Update controller weights $\Phi$

---

At a high-level, the computational complexity of algorithm 3 is $\mathcal{O}(E \times B \times L \times F)$, where $E$ is the number of search iterations, $B$ is the number of trees constructed per iteration, $L$ is the number of loss-minimization steps, and $F$ is the cost of a single forward pass of a tree. The $B$ trees are optimized independently, allowing straightforward parallelization to reduce runtime. The cost $F$ depends mainly on the number of weight parameters $\boldsymbol{\theta}$, not on the number of operators $\mathbf{e}$ in the tree. Looking at fig. 7, we see that adding extra nodes has a minor impact on computational cost, as it merely requires additional inexpensive evaluations of the unary, binary, or interation operators.

## C  Additional Experiment Results

We show the H-FEX tree structures in Figure 7 used for each of the numerical experiments in Section 4. The tree structure is fixed before training, and the operators are the final learned operators. Weights are added after each of the unary and binary nodes.

We report the average runtimes of H-FEX and other methods across 10 independent runs for the non-separable system and the three-body problem in table 1. SINDy is notably fast, as its core computation is solving a single linear regression. H-FEX runs slower on the three-body problem than in the non-separable system, due to larger weight matrices shown in fig. 7. Currently, the score calculation in H-FEX is implemented in serial, but several components are naturally parallelizable. In particular, score estimation for operator sequences within a batch and the fine-tuning of candidate models can be parallelized to improve efficiency in larger-scale settings.

We plot in fig. 8 the empirical probability of H-FEX having the correct expression in the pool as a function of the number of search loop iterations. As the number of iterations increase, H-FEX's success rate increases. In both experiments, we run for 100 iterations. H-FEX finds the correct expression in 9 out of 10 runs for the nonseparable system and in 8 out of 10 runs for the three-body problem.

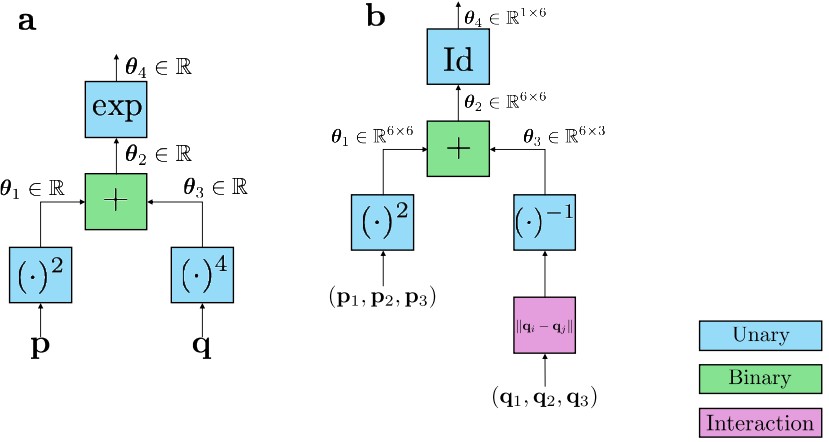

Figure 7: **Tree structure of H-FEX for numerical experiments**. **(a)** shows the tree structure and final learned operator sequence of H-FEX for the nonseparable experiment in Section 4.1. **(b)** shows the tree and final learned operators, where the weights $\boldsymbol{\theta}_1, \ldots, \boldsymbol{\theta}_4$ are optimizable parameters of the form $\mathbb{R}^{d_{\text{out}} \times d_{\text{in}}}$.

| Method | Non-separable system | Three-body problem |
|--------|:---:|:---:|
| H-FEX | 1.62 hrs | 4.54 hrs |
| SINDy | <5 secs | - |
| SRNN | - | 3.86 hrs |
| HNN | - | 1.14 hrs |
| SANN | - | 15.01 hrs |

Table 1: **Average runtime of each method** on two benchmark problems, computed over 10 independent runs. For the three-body problem, runtimes are based on training with trajectories from the interval $[0, 7]$.

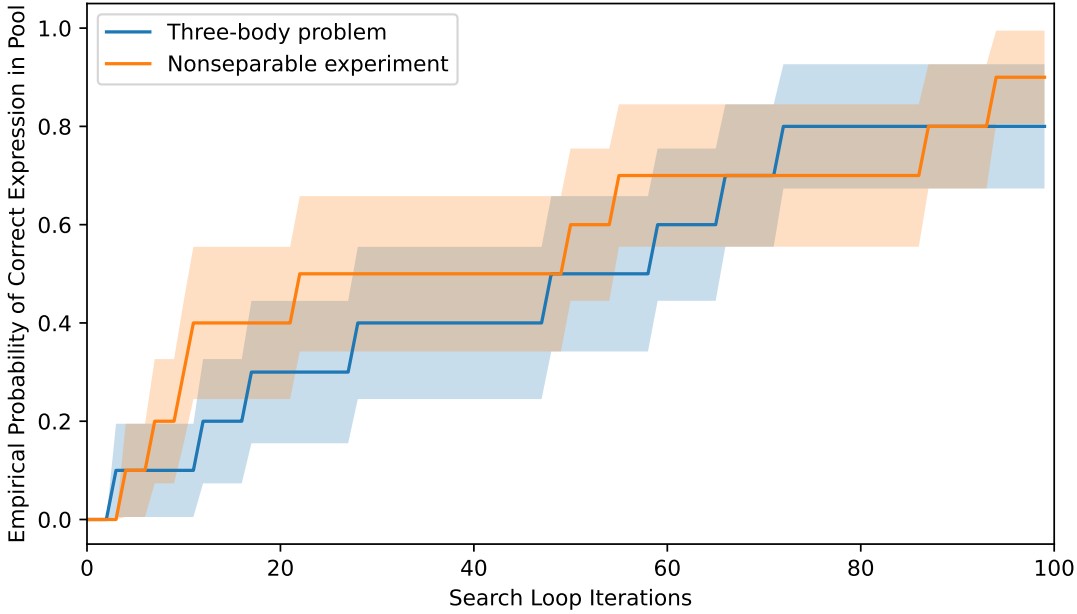

Figure 8: **Empirical probability of H-FEX adding the correct expression to the pool** for two benchmark problems. For each problem, we conducted 10 independent runs, recording whether H-FEX successfully identified the correct expression and the iteration in which it was first added to the pool. Shaded regions indicate $\pm 1$ standard error of the probability estimate.

Additionally, we demonstrate that H-FEX can still recover the true Hamiltonian expression when using an alternative tree structure. By including the identity operator in the dictionaries of interaction nodes, the H-FEX tree can effectively reduce to an equation represented by a subtree. As shown in fig. 9 and fig. 7 (b), two distinct tree structures yield the same final expression. In general, larger H-FEX trees can produce expressions functionally equivalent to smaller trees by leveraging identity operators or by learning weights that are close to zero.

## D  Ablation Studies

We test H-FEX with and without interaction nodes on the three-body problem in fig. 10. While both variants had similar scores, the version without interaction nodes generalizes poorly, as it fails to find an expression capable of modeling the multi-body coordinates in eq. (16).

We also conduct an ablation study to understand the effect of the integration method used in the H-FEX search loop in table 2. All experiments use the integrator in eq. (11) during training. Replacing it with eq. (10) results in a substantial drop in the average loss and score of H-FEX trees in the pool.

## E  H-FEX on Real-World Astronomical Data

We apply the proposed H-FEX method to real-world astronomical data from the Jupiter–Sun system, which can be approximated as a two-body problem. While the governing equations for this system are well-understood, this experiment illustrates how H-FEX can verify a proposed symbolic form and assess whether alternative forms provide a better fit, an approach that can also be applied to systems with unknown dynamics. The dataset is obtained from the *JPL Horizons OnLine Ephemeris System*(`https://ssd.jpl.nasa.gov/horizons/`), an online service provided by NASA's Jet Propulsion Laboratory that generates

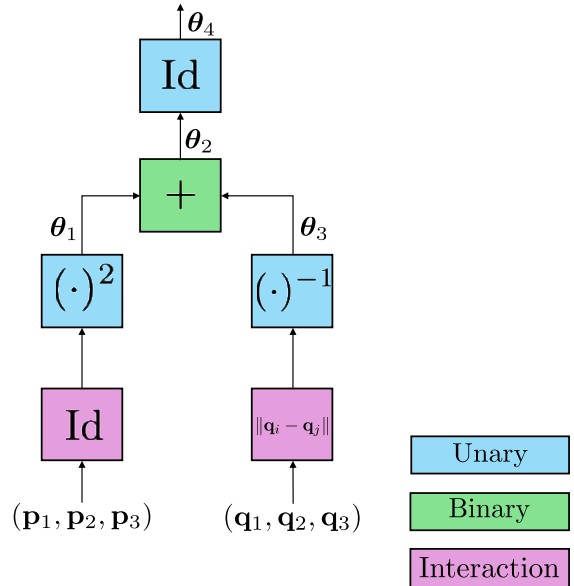

Figure 9: **Alternative tree structure for three-body problem.** The operator dictionaries used are $\mathbb{U} = \{\text{Id}, (\cdot)^2, (\cdot)^3, \exp, \sin, (\cdot)^{-1}\}$, $\mathbb{B} = \{+, \times, -\}$, and $\mathbb{I} = \{\text{Id}, \|\mathbf{q}_i - \mathbf{q}_j\|^2, \|\mathbf{q}_i - \mathbf{q}_j\|, \|\mathbf{q}_i \odot \mathbf{q}_j\|^2, \|\mathbf{q}_i \odot \mathbf{q}_j\|\}$. H-FEX hyperparameters are the same as those mentioned in section 4.2. This tree reduces to the same expression in fig. 7 (b).

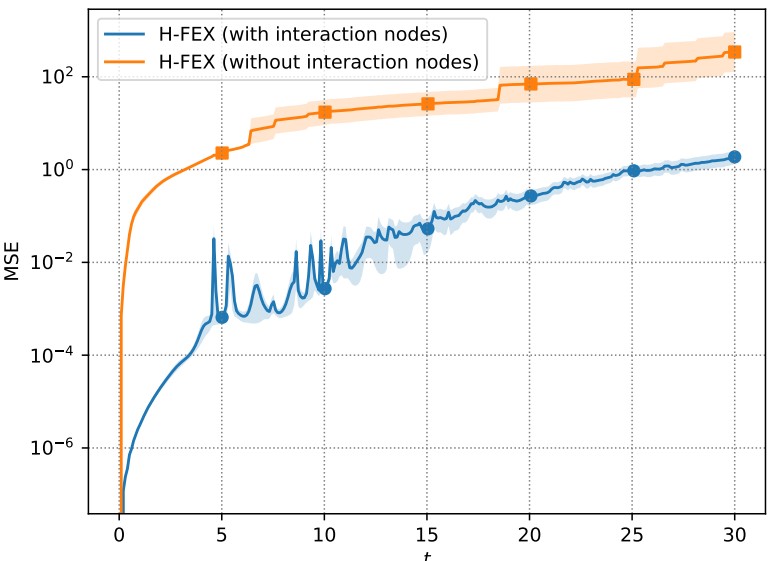

Figure 10: **Comparing MSE over time for H-FEX with and without interaction nodes** on the three-body problem trained on trajectories from $[0, 7]$. Without interaction nodes, predicted trajectories diverge rapidly from the ground truth.

| Integration method | Loss | Score |
|---|---|---|
| Integrating with eq. (10) | $0.8800 \pm 0.1115$ | $0.5338 \pm 0.0335$ |
| Integrating with eq. (11) | $\mathbf{0.0046 \pm 0.0015}$ | $\mathbf{0.9954 \pm 0.0015}$ |

Table 2: **Ablation study comparing integration methods** on the three-body problem trained on trajectories from $[0, 7]$. The table reports the mean with one standard deviation for the loss and score of final expressions in the pool after the H-FEX search loop. The integration method in eq. (11) leads to better loss and score by using the data to prevent the propogation of integration errors.

accurate ephemerides for solar system bodies. We set the target body to be the *Jupiter Barycenter*, the coordinate center to be the *Solar System Barycenter*, and the time span from 2021-01-01 to 2025-01-01 with a time step of 1 day, yielding a total of 1462 samples. The original data is provided in kilometers (for position) and kilometers per second (for velocity). The momentum data is computed as $\mathbf{p} = m_J \mathbf{v}$, where $m_J$ is the mass of Jupiter.

Roughly, the Newtonian two-body Hamiltonian for the Sun–Jupiter system in the center-of-mass frame is

$$\mathcal{H}(\mathbf{p}, \mathbf{r}) = \frac{\|\mathbf{p}\|^2}{2m_J} - \frac{Gm_S m_J}{\|\mathbf{r}\|}, \tag{18}$$

where $\mathbf{r} \in \mathbb{R}^3$ is the relative position (Jupiter relative to Sun), $\mathbf{p} \in \mathbb{R}^3$ is momentum, and $G \in \mathbb{R}$ is the gravitational constant. For numerical stability and interpretability we nondimensionalize using the canonical solar-system scales

$$L_0 = 1 \text{ AU} = 1.496 \times 10^{11} \text{ m}, \quad M_0 = m_S, \quad T_0 = 1 \text{ yr} = 365.25 \times 86400 \text{ s}.$$

We define dimensionless variables

$$\tilde{\mathbf{r}} = \frac{\mathbf{r}}{L_0}, \qquad \tilde{\mathbf{p}} = \mathbf{p} \frac{T_0}{M_0 L_0}, \qquad \tilde{t} = \frac{t}{T_0}, \qquad \tilde{m}_J = \frac{m_J}{M_0}.$$

Under these choices, the gravitational constant rescales to

$$\tilde{G} = \frac{GM_0 T_0^2}{L_0^3} \approx 4\pi^2,$$

and the dimensionless reduced Hamiltonian becomes

$$\tilde{\mathcal{H}}(\tilde{\mathbf{p}}, \tilde{\mathbf{r}}) = \frac{\|\tilde{\mathbf{p}}\|^2}{2\tilde{m}_J} - \frac{\tilde{G}\tilde{m}_J}{\|\tilde{\mathbf{r}}\|}. \tag{19}$$

The sampling interval $\Delta t_{\text{phys}} = 1$ day corresponds to

$$\Delta t = \frac{1 \text{ day}}{1 \text{ yr}} = \frac{1}{365.25} \approx 0.00273785.$$

Substituting $m_J = 1.90 \times 10^{27}$ kg, $m_S = 1.99 \times 10^{30}$ kg in Eqn. (19), we have

$$\tilde{\mathcal{H}}(\tilde{\mathbf{p}}, \tilde{\mathbf{r}}) = 523.56\|\tilde{\mathbf{p}}\|^2 - \frac{3.77 \times 10^{-2}}{\|\tilde{\mathbf{r}}\|}. \tag{20}$$

We use an H-FEX tree structure similar to (b) of fig. 7, but with two bodies instead of three. The relative coordinates of the dimensionless dataset can be converted into absolute coordinates by fixing one body at the origin. For training, we use the same hyperparameters as those reported in for the three-body problem in section 4.2, and after the H-FEX search loop has concluded, we fine-tuned models in the pool for $5,000$ iterations using Adam ($\beta_1 = 0.9, \beta_2 = 0.999$ with a constant learning rate of $10^{-1}$).

After fine-tuning, we obtained a model with a loss of $1.1657 \times 10^{-9}$ which yielded the expression:

$$\hat{\mathcal{H}}(\tilde{\boldsymbol{p}}, \tilde{\boldsymbol{r}}) = 523.5652\tilde{p}_x^2 + 523.5655\tilde{p}_y^2 + 468.8063\tilde{p}_z^2 + \frac{-0.0412}{\|\tilde{\boldsymbol{r}}\|},$$

where $\tilde{\boldsymbol{p}} = (\tilde{p}_x, \tilde{p}_y, \tilde{p}_z)^\top$ is the momentum on the xyz axis. Comparing with eq. (20), our results show that the learned coefficients are in close agreement with the true values with exception of the z-component, demonstrating that H-FEX is capable of verifying that the two-body model is a good fit while testing for potentially better alternative forms.

