# OpenReview forum: "H-FEX: A Symbolic Learning Method for Hamiltonian Systems"
_TMLR — Rejected by TMLR_

### Review · Reviewer_8mN2 · 2025-07-30

**Summary Of Contributions:**

This paper proposes a method for learning a closed-form expression of Hamiltonian systems. The approach represents mathematical operators as nodes in a tree, with trainable weights assigned to each node. The learning process is framed as a reinforcement learning problem and optimized via policy gradients. The method is evaluated on two benchmark systems with known closed-form solutions, achieving state-of-the-art performance.

**Audience:**

Yes

**Audience Explanation:**

Yes. Researchers interested in symbolic regression, interpretable physics modeling, or learning closed-form expressions for Hamiltonian systems would likely find this work relevant. However, it is currently unclear whether the method requires prior knowledge of the general structure of the system to succeed, which could limit its practical applicability. Given this uncertainty, and in accordance with the TMLR acceptance criteria, I mark this as a "yes."

**Claims And Evidence:**

Yes

**Claims Explanation:**

The paper's claims are convincingly supported for systems where the initial tree structure aligns well with the ground truth dynamics. In these cases, the method achieves strong results. However, it remains unclear how well the approach generalizes to more complex or mismatched systems. See my questions in the "Requested Changes" section for further clarification

**Requested Changes:**

My main concern is that the paper focuses on simple systems where the initial tree structure is already well aligned with the true dynamics. This raises concerns about inductive bias and generalizability. I list below several questions, sorted by priority (high to low):


(1) How does the method perform on more complex systems, such as the MuJoCo environments used in [1]? Without this, it is difficult to assess whether the method genuinely outperforms others or simply benefits from a stronger inductive bias on the selected benchmarks.

(2) Is the structure of the tree fixed during training, or can it be adapted — e.g., by pruning or adding nodes?

(3) Can the model learn dynamics from raw pixel input, similar to [2]?

(4) Why is Eq. (11) used for training instead of Eq. (10)? An ablation study comparing the two would clarify the choice.


Addressing question (1) is essential for determining the broader applicability of this approach. I believe it must be answered for the paper to be considered for acceptance.


[1] Gruver et al, Deconstructing the Inductive Biases of Hamiltonian Neural Networks

[2] Greydanus, Hamiltonian Neural Networks

---

> ### Author Response · Authors · 2025-08-17
> **Response to Reviewer 8mN2**
>
> Thank you for careful review of our submission. We provide point-by-point responses to each comment below. All revisions are highlighted in red in the paper.
>
> > **Q1:** MuJoCo environments and concerns about inductive bias
>
> We thank the reviewer for suggesting the use of MuJoCo environments. While these environments are widely used benchmarks in reinforcement learning, they are not well aligned with the current scope of our work. Specifically, MuJoCo systems are generally non-conservative due to effects such as friction and collisions with the environment, whereas H-FEX is designed for conservative dynamical systems. Instead, to demonstrate applicability beyond synthetic benchmarks, we also evaluate H-FEX on real-world astronomical trajectory data in Section E of Appendix, which provides a practical and physically meaningful testbed consistent with our problem formulation.
>
> Regarding the reviewer’s concern about inductive bias, we note that H-FEX is a symbolic regression method, whose goal is to recover interpretable closed-form Hamiltonians. Symbolic regression methods, including SINDy, inherently involve some inductive bias through their representation structure, which constrains the search to interpretable forms. In H-FEX, this structure is provided by the tree representation and candidate operator dictionary, but these choices define a flexible search space that can accommodate a wide range of Hamiltonian systems. This differs from black-box approaches such as HNN, which can approximate the Hamiltonian but cannot provide an explicit formula.
>
> > **Q2:**  Is the tree fixed, or can it be adaptive?
>
> We thank the reviewer for raising this question. In all of our current experiments, the tree structures of the unary, binary, and interaction nodes are fixed. As noted in the conclusion, adapting the tree structure is an interesting direction for future work, and we have not explored adaptive trees in this paper. One possible approach is to start with a larger tree and allow the controller to prune nodes with small coefficients to reduce the tree size. Another approach is to configure the controller to output probability distributions over whether to include a node at a specific position to get a more flexible, adaptive structure.
>
>
> > **Q3:**  Can H-FEX learn dynamics from raw pixel input similar to HNNs?
>
> The paper [2] uses an autoencoder framework with HNN to learn the dynamics from raw pixel input. The encoder transforms raw pixel input to a latent space where an HNN can learn the dynamics, and the decoder transforms the HNN’s outputs back to pixel space allowing them to compute the loss. The same setup could be used with H-FEX, but the interpretability of the closed-form expressions would be lost, as the output of the encoder is not interpretable. To enable H-FEX to learn from pixel inputs in a meaningful way, it would require transforming the pixel inputs to interpretable coordinates, such as position and momentum.
>
> [2] Greydanus, Hamiltonian Neural Networks
>
> > **Q4:**  Why eqn (11) is used for training over eqn (10)
>
> We use eqn (11) for integration during training since it prevents the accumulation of integration errors through the trajectory, which can facilitate the training. Consider the scenario where our dataset consists of three points: $(p_{t_0}, q_{t_0}), (p_{t_1}, q_{t_1}), (p_{t_2}, q_{t_2})$. Equation (10) uses the data $(p_{t_0}, q_{t_0})$ to get prediction $(p_{t_1}', q_{t_1}')$, and then uses prediction $(p_{t_1}', q_{t_1}')$ to get prediction $p_{t_2}', q_{t_2}')$. If the prediction $(p_{t_1}',q_{t_1}')$ was not accurate, then this error propogates to the next prediction $(p_{t_2}', q_{t_2}')$. In equation (11), we use the available data $(p_{t_1}, q_{t_1})$ as the initial condition to get prediction $(p_{t_2}', q_{t_2}')$.
>
> This effect becomes more pronounced in stiff dynamical systems. We further provide an ablation study comparing these integration methods in table 2 of the revision. From table 2, we see that optimizing using the integrator in eqn $(11)$ yields a much smaller loss, resulting in larger scoring proposals by H-FEX.

---

> > ### Comment · Reviewer_8mN2 · 2025-08-19
> >
> > I thank the authors for their thorough rebuttal and for adding additional experiments.
> >
> > On Q1 (MuJoCo/generalization): I acknowledge the authors’ point that MuJoCo environments are non-conservative and thus not directly aligned with H-FEX. However, this does not fully resolve my concern. My main point was that the paper only evaluates on systems where the representation aligns closely with the true dynamics. The added astronomical data is interesting, but it still falls within the same class of well-structured conservative systems. Without evaluation on more complex or mismatched settings, it remains unclear whether H-FEX is genuinely robust or primarily succeeds due to favorable inductive bias. I continue to see this as a significant limitation.
> >
> > On Q2 (tree adaptivity): I appreciate the clarification, though I view fixed tree structures as a notable constraint that should be emphasized more clearly as a current limitation.
> >
> > On Q3 (raw pixels): The explanation is reasonable, but it reinforces my impression that H-FEX is currently limited to settings where structured inputs (e.g., positions, momenta) are already available. This further narrows its scope compared to related approaches.
> >
> > On Q4 (Eq. 11 vs. Eq. 10): Thank you for your explanation.

---

> > > ### Author Response · Authors · 2025-08-19
> > > **Follow-up response to Reviewer 8mN2**
> > >
> > > We thank the reviewer for the follow-up comments.
> > >
> > > We agree that symbolic regression–based methods such as H-FEX might be less flexible than purely neural network–based approaches in settings for MuJoCo or with raw pixel inputs, since they require predefined operator dictionaries and certain assumptions about the solution structure. However, this tradeoff also provides unique advantages: once an appropriate representation is identified, symbolic regression can yield interpretable models and stable long-term predictions, whereas neural networks often suffer from trajectory divergence over time. In this sense, H-FEX is not intended as a universal solution but as a complementary approach that specifically targets interpretable modeling of Hamiltonian systems.
> > >
> > > We also experimented with alternative symbolic regression–based methods such as ODEformer, as mentioned by Reviewer 5KYR. However, ODEformer struggled to obtain solutions that were close to the true dynamics for the non-separable case in Section 4.1, which highlights the difficulty of directly applying generic symbolic regression to Hamiltonian systems. By contrast, H-FEX leverages domain knowledge tailored to Hamiltonian functions, which makes it more effective for these problems despite its limitations.

---

### Review · Reviewer_5KYR · 2025-08-03

**Summary Of Contributions:**

This paper proposes a symbolic learning method named H-FEX, which aims to discover the governing equations of Hamiltonian systems from observational data. The method enforces physical constraints and effectively captures complex interactions in many-body systems by introducing an interaction node mechanism customized for Hamiltonian systems. Through experiments on a non-separable Hamiltonian system and a three-body problem, the authors demonstrate that H-FEX can recover interpretable Hamiltonian functions. Long-term numerical simulations based on the discovered functions show superiority in trajectory prediction accuracy (MSE) and energy conservation compared to traditional symbolic regression methods (SINDy) and advanced black-box neural network models (HNN, SRNN, SANN).

#### Key Strength:

 - The method's design inherently ensures the conservation of system energy by directly modeling the Hamiltonian structure.

#### Key Weaknesses:

 - The paper lacks a compelling real-world application, as its main test case uses a problem with a known Hamiltonian.
 - The comparison to baselines is insufficient, omitting several state-of-the-art methods.
 - The evaluation lacks a key metric for symbolic regression: the formula recovery success rate.

**Additional Comments:**

To support my review, I would like to provide a more detailed elaboration on the key strengths and weaknesses of this submission, along with the full references for the papers cited in my comments.

#### Strength:

 - #### Physical Conservation

   By targeting the Hamiltonian dynamical structure in symbolic regression and then substituting the expression into Hamilton's equations to drive the system's evolution, H-FEX ensures the conservation of system energy by design. The extremely low "relative energy error" in the experiments validates this point.

#### Weaknesses:

 - #### Lack of Valuable Real-World Application Scenarios

   The paper's core objective is to use symbolic regression to solve for unknown Hamiltonians, but it fails to provide a real-world case where the Hamiltonian is unknown and its discovery would have significant application value. The core validation case used in the paper—the three-body problem—has a known Hamiltonian; the real difficulty lies in solving its trajectory, not discovering its governing equations. This makes the method seem more like a test on a benchmark problem with a known answer, limiting its practical utility.
 - #### Insufficient Baseline Comparison

   The baselines compared in the paper are inadequate. For instance, in the first experiment, the authors only compare with SINDy, neglecting more advanced SOTA methods in the field of symbolic regression, including the Transformer-based ODEFormer [1], the physics-informed AI Feynman [2] and Φ-SO [3], and the LLM-based ICSR [4]. The lack of comparison with these methods makes it impossible for readers to judge the true performance of H-FEX.
 - #### Absence of Key Evaluation Metrics

   The paper avoids a core evaluation metric for symbolic regression tasks: the formula recovery success rate. The paper only presents the optimal formula from a single run and does not report to what extent H-FEX can recover the Hamiltonian function, making it difficult to assess the method's stability and reliability. Relying solely on indirect metrics from numerical simulations (like MSE) is insufficient to evaluate the performance of a symbolic regression method.



**References**

[1] d'Ascoli, S., et al. (2024). ODEFormer: Symbolic Regression of Dynamical Systems with Transformers. In *International Conference on Learning Representations*.

[2] Udrescu, S. M., & Tegmark, M. (2020). AI Feynman: A physics-inspired method for symbolic regression. *Science Advances*, 6(16), eaay2631.

[3] Tenachi, W., Ibata, R., & Diakogiannis, F. I. (2023). Deep symbolic regression for physics guided by units constraints: toward the automated discovery of physical laws. *The Astrophysical Journal*, 959(2), 99.

[4] Merler, M., Haitsiukevich, K., Dainese, N., & Marttinen, P. (2024). In-context symbolic regression: Leveraging large language models for function discovery. *arXiv preprint arXiv:2404.19094*.

**Audience:**

Yes

**Audience Explanation:**

Yes, the general problem of discovering governing equations of physical systems from data is a topic of significant interest at the intersection of machine learning and various scientific disciplines. There is a substantial audience, particularly within the sub-fields of symbolic regression and physics-informed ML, for methods that can automatically produce interpretable physical laws, and this interest is not limited to just Hamiltonian systems.

However, while the general topic is of high interest, the specific framing and execution of this paper may limit the appeal of its findings. The paper does not present a compelling, real-world scenario where a system's Hamiltonian is truly unknown and its discovery would be critical. By focusing on a problem with a well-established Hamiltonian (the three-body problem ), the work feels more like an academic benchmark than a practical discovery tool, which may temper the enthusiasm of a broader audience.

**Claims And Evidence:**

No

**Claims Explanation:**

The claims of the paper, particularly regarding the superiority and reliability of the H-FEX method, are not supported by sufficiently convincing evidence due to two main weaknesses in the experimental evaluation:

1. **Insufficient Baseline Comparison:** The paper claims superiority over other methods , but the evidence is unconvincing because the comparison is inadequate. It fails to include several modern SOTA methods in symbolic regression (e.g., ODEFormer [1], AI Feynman [2]). Without these crucial comparisons, the evidence for H-FEX's superiority is weak, as its performance against the true state of the art is unknown.
2. **Absence of Key Evaluation Metrics:** The paper claims that H-FEX can recover the true Hamiltonian function, but it omits a core metric for this task: the formula recovery success rate over multiple runs. This metric is essential for evaluating the stability and reliability of a symbolic regression method. By only presenting a single best-case result and relying on indirect metrics like trajectory MSE, the paper does not provide convincing evidence that the method can reliably discover the correct governing equations.

**Requested Changes:**

Here is a list of requested changes. The first three are considered **critical** for the paper to reach the standard for acceptance. The fourth is a suggestion that would **strengthen** the paper's contribution.

1. **Expand Baseline Comparisons (Critical):** To substantiate the claims of superiority, the paper must include experimental comparisons against more current and relevant state-of-the-art symbolic regression methods. This should include, but is not limited to, methods like ODEFormer [1] and the physics-informed AI Feynman [2]. These comparisons should be conducted on the same experimental setups presented in the paper to provide a fair assessment of H-FEX's performance.
2. **Include Standard Evaluation Metrics (Critical):** The evaluation must be strengthened by including standard metrics for symbolic regression tasks. Specifically, the authors should report the **"formula recovery success rate"** over multiple independent runs. This is essential to demonstrate the method's stability and reliability.
3. **Introduce a More Compelling Experimental Scenario (Critical):** To demonstrate practical utility, the motivation for the work must be significantly strengthened by introducing a new experimental scenario. This scenario should be a real-world case or, at a minimum, a problem where the governing Hamiltonian is not as mature as in the current examples. The goal of this addition is to present a situation where discovering the Hamiltonian is in itself an important and non-trivial task, thereby validating the practical relevance of the H-FEX method.
4. **Ablation Study on the "Interaction Node" (Strengthening):** The "interaction node" is presented as a key technical contribution for handling multi-body systems. To provide direct evidence of its necessity and effectiveness, an ablation study should be conducted. Specifically, for the three-body problem, the authors should perform an experiment running H-FEX without the specialized interaction operators in the dictionary. This would rigorously validate the importance of this novel component.

---

> ### Author Response · Authors · 2025-08-17
> **Response to Reviewer 5KYR**
>
> Thank you for careful review of our submission. We provide point-by-point responses to each comment below. All revisions are highlighted in red in the paper.
>
> > **Q1**: Expand baseline comparisons
>
> We thank reviewer for the suggestion additional comparisons with four different state-of-the-art symbolic regression methods: ODEFormer, AI Feynman, $\Phi$ -SO, and ICSR. These methods represent significant advances, but each is developed for a setting that differs from the one considered in this work.
>
> First, AI Feynman, $\Phi$ -SO, and ICSR are general symbolic regression methods that solve problems of the form $y=f(\mathbf{x})$. They give a closed-form expression for $f$ but require data in the form $(\mathbf{x}, y)$. This makes it difficult to directly apply these methods to learn the hamiltonian function $\mathcal{H}(p, q)$ given a dataset of trajectory values $(p(t), q(t))$, since we do not have any output values of $\mathcal{H}$. Even if the task were for these methods to learn the governing equations $(dp/dt, dq/dt)$, this would not be simple, as they were not primarily designed to learn dynamical systems. One issue here is that these methods may not be able to jointly fit multiple output equations.
>
> Second, ODEFormer is designed for symbolic regression of dynamical systems and are capable of giving expressions for $(dp/dt, dq/dt)$, but it can only handle one single trajectory $(p(t), q(t))$ as the input. This is a weakness that is stated in [1]. Our problem consists of using multiple trajectories to learn a single expression, and ODEFormer is only capable of giving a separate expression for each individual trajectory. We also made an effort to evaluate ODEFormer using its official pre-trained model. Specifically, we attempted to fit the non-separable case in our paper with a single trajectory. However, the predicted $(dp/dt, dq/dt)$ diverged considerably from the ground truth. Since it is unclear whether this discrepancy arises from limitations of the pre-trained model or other factors, we do not report the result in detail here.
>
> In light of these considerations, we would like to point out that these methods are not directly applicable to the problem setup to learn the Hamiltonian, and we have clarified the above reasons in the revised manuscript in Introduction.
>
> [1] d'Ascoli, S., et al. (2024). ODEFormer: Symbolic Regression of Dynamical Systems with Transformers. In International Conference on Learning Representations.
>
> > **Q2**: Evaluation metrics
>
> The reviewer suggests adding a formula recovery success rate over multiple independent runs. In the revision, we have added figure 8 in the appendix to show the formula success rate with respect to iterations of the search loop. We added this figure because the formula recovery success rate is usually dependent on the number of iterations used, with more iterations seeing a higher success rate.
>
> > **Q3**: A more compelling experimental scenario
>
> We thank reviewer's suggestion on a real-world case to demonstrate the practical relevance of H-FEX. For this we have added section E in the appendix, where we apply H-FEX to real-world astronomical data of Jupiter and the Sun. We use H-FEX to validate that the Jupiter-Sun system can indeed be approximated using a two-body system. In this case, we already know certain parameters in the Hamiltonian function, so we can validate the correctness of H-FEX. Furthermore, a similar application of H-FEX could be used to verify or test hypothesized models for less-understood celestial bodies. The tree structure of H-FEX could then be set to represent the hypothesized models.
>
> > **Q4**: Ablation study on the Interaction Node
>
> We thank the reviewer for suggesting an ablation study to provide direct evidence of the interaction node’s necessity. We have added this to a paragraph in section D of the appendix and in figure 9 to test H-FEX with and without interaction nodes on the three-body problem. From figure 9, we see that H-FEX with interaction nodes provides a much better model of the system compared to H-FEX without interaction nodes.

---

### Review · Reviewer_2oRc · 2025-08-04

**Summary Of Contributions:**

A Hamiltonian system is a set of ordinary differential equations characterized by a Hamiltonian function.
This paper proposes H-FEX (Finite Expression Method for learning Hamiltonian Systems), a method that can learn a symbolic Hamiltonian function of a Hamiltonian system, given trajectories from the system as training set.
H-FEX has more expressive power compared with existing interpretable methods (SINDy), and better long-term robustness compared with neural-based methods (HNN, SRNN, SANN).
H-FEX is based on FEX (Finite Expression Method) and is specifically designed for Hamiltonian systems.

* Strengths

The presentation is very easy to follow. The motivation of designing H-FEX is clear and strong. The experiments are generally convincing and demonstrate the interpretability, expressive accuracy and long-term robustness of H-FEX.

* Weaknesses

1. H-FEX highly relies on the prior knowledge of the tree structure and the candidate operator dictionaries.
2. Since H-FEX requires proposing candidate operator dictionaries and evaluating them one by one, it is possible that H-FEX may not be computationally efficient to systems with larger tree structures. All experiments have tree size less or equal than $5$.
3. See below questions.

* Questions

1. In step 2 of Section 3.1, the definition of $S(e)$: In principle, choosing $S(e) = - L(e)$ suffices to order different operator sequence $e$. Do the authors have more insight on why choose $S(e)$ in such a specific form? ($S(e) = 1 / (1 + L(e))$)
2. In Figure 2 in page 6 and Figure 7 in Appendix C: The interaction node that acts on $(q _1,q _2,..., q _n)$ will have $n(n-1)/2$ scalar outputs, and the unary node $(\theta _1)$ that acts on $(p _1, ...,p _n)$ will have $n$ scalar outputs. How can the binary operator $\theta_2$ act with different number of inputs (when $n \ne 3$)? How can the tree outputs a single scalar for general $n$?
3. How much time (or estimated computational resources) does it take in two experiments for H-FEX and baselines? Please clarify this in main body.
4. In page 7, last line: What is the $\beta _1, \beta _2$ hyper-parameters in Adam optimizer? Please clarify this in main body.
5. In page 8, first equation: The precision does not unify in $\hat{\mathcal{H}} _{\Theta}(p,q)$. Please increase the precision of $1.000$ (as well as following experiments).
6. In page 9, second paragraph: What is the time step in three-body experiment? Is it still 0.1 as in the first experiment? Please clarify this in main body.
7. In Figure 4: Since the SAI-axis is on logarithm scale, what is the density computation approach? Computing the density of SAI first and then taking logarithm on SAI-axis, or taking logarithm on SAI and then computing the density of log(SAI)?
Please clarify that the density function is on SAI or log(SAI).
8. In Figure 5: Why the authors do not apply SANN on this experiments? What are the difficulties the authors face?
9.  Comparing Figure 5 and Figure 7: In my understanding, if the time step is fixed 0.1 in both [0,3] and [0,7] dataset, then the first dataset is a proper subset of the second dataset. Then, training on the first, smaller dataset is expected to yield worse performance than training on the second, larger dataset. However, the experiments indicate the opposite for HNN and SRNN. Can the authors explain the insight behind this phenomenon?

**Additional Comments:**

The paper might have been de-anonymized by *Acknowledgement* section.

**Audience:**

Yes

**Audience Explanation:**

The paper claims to advance the learning of Hamiltonian systems, which might attract researchers in learning theory, AI for Science, etc.

**Broader Impact Concerns:**

I think this paper has no ethical concerns.

**Claims And Evidence:**

Yes

**Claims Explanation:**

The paper executes two experiments to show the effectiveness of H-FEX. In both experiments, the performance of H-FEX surpasses all baselines. Most of the experimental details are clearly provided, making the experiments reproducible.

**Requested Changes:**

* page 3: In practise -> In practice
* See comments in my questions above.

---

> ### Author Response · Authors · 2025-08-17
> **Response to Reviewer 2oRc**
>
> Thank you for careful review of our submission. All revisions are highlighted in red in the paper.
>
> > **W1** H-FEX highly relies on the prior knowledge
>
> As a symbolic regression method, H-FEX requires a representation structure to constrain the search space, which is a common feature of symbolic regression approaches such as SINDy. This inductive bias does not prescribe the true dynamics but instead defines a flexible search space in which candidate models can be explored. The effectiveness of this framework is further demonstrated by our experiments on real-world astronomical trajectory data (Appendix E in the revision), where H-FEX successfully identifies Hamiltonians consistent with the system.
>
> > **W2** Computationally efficiency
>
> We agree that the search phase in H-FEX introduces computational overhead, as it involves both RL-based optimization of the controller and the evaluation of operator sequence scores. However, this design also provides flexibility, since users can adjust hyperparameters, such as the number of RL iterations, the number of score-function evaluations, the tree depth, and the batch size, to balance efficiency and performance for a given problem or resource budget.
>
> As reported in Table 1 and Appendix C (response to **Q3**), the runtime of H-FEX is higher than that of NN-based methods. Currently, the score calculation and fine-tuning in H-FEX are implemented in serial, but several components are naturally parallelizable. In particular, score estimation for operator sequences within a batch and the fine-tuning of candidate expressions can be parallelized to improve efficiency.
>
> > **Q1** Insight to definition of $S(e)$
>
> We agree that there exists many different ways to define $S(e)$, such as $S(e):=-L(e)$ as mentioned by the reviwer, that do not impact the ranking of different operator sequence $e$. However, choosing $S(e):= 1/(1 + L(e))$ prevents large updates to the controller’s weights, which can keep the training stable even when the losses vary wildly. Using $S(e):= 1/(1 + L(e))$, bounds the score, and it prevents the term  $S(e^{(j)})$ in eqn (9) from becoming large. Using $S(e):=-L(e)$, it is possible for the term $S(e^{(j)})$ to become large if you had one operator sequence with small loss and many operator sequences with large losses. We have added this insight in the revision at the bottom of page 4.
>
>
> > **Q2** How a tree outputs a scalar for general n
>
> Indeed, the output of the interaction node will have $n(n-1)/2$ scalar outputs and may not match the output of a sibling node. However, after applying weight parameters $\theta_1$ and $\theta_3$, we can force the incoming values to agree in dimension before processing by the binary operator and $theta_2$ subsequently. There are many ways to achieve this, but we show how we specifically configured the weights in figure 7(b) of the revision. In general, the “rules” for a valid tree to output a scalar for input of size n requires:
> 1. transforming vectors to scalars somewhere in the tree,
> 2. the incoming inputs to binary nodes must be the same.
>
>
> > **Q3** Runtime
>
> We reported the runtimes for all our experiments in table 1 of the revision. We additionally provide an analysis of the computational complexity of H-FEX at the bottom of page 15 in the appendix.
>
> > **Q4** Adam hyperparams
>
> In all our experiments, we use Adam with $\beta_1=0.9$ and $\beta_2=0.999$. This information has been added in the revision.
>
> > **Q5** Consistent precisions
>
> We change the precision to 4 digits for all reported parameter values on page 8.
>
> > **Q6** Timestep of datasets
>
> In revision, we clarify that the timesteps for all datasets in the three-body problem is $0.1$ on page 9.
>
> > **Q7** SAI clarification
>
> In figure 4, we compute the density of SAI first and then plot it on log scale. This has been clarified on the bottom of page 9 and figure 4.
>
> > **Q9** Why the $[0, 7]$ dataset is more difficult than $[0, 3]$ dataset
>
> While the $[0, 7]$ dataset contains the $[0, 3]$ dataset, it poses additional challenges because the system begins to exhibit singularities and stiff components in the dynamics. In the three-body problem, collisions lead to such behavior, which arises when $q_i - q_j$ is small in eqn. (16). In the $[0, 3]$ dataset, the system typically has not evolved long enough for collisions to occur, whereas in the $[0, 7]$ dataset, collisions become more likely. This is reflected in fig. 4, where the $[0, 7]$ dataset includes time points with large SAI values. Although regions with low SAI also exist, it is the presence of large SAI values that highlights the need for the surrogate to accurately characterize the singular and stiff components of the dynamics.
>
> > **Q8** Why no SANN in fig 5
>
> We choose to omit SANN from the three-body problem with the $[0, 3]$ dataset (fig 5) due to the lack of stiff regions established in the previous point. SANN is a method designed for handling stiff systems, so we only apply SANN to the $[0, 7]$ dataset.

---

> > ### Comment · Reviewer_2oRc · 2025-08-20
> > **Response to Authors**
> >
> > Thank you for your response and revision of your submission.
> >
> > W1: This answer is partially satisfactory. One remaining concern is (from Reviewer 8mN2) that the effectiveness is possibly a coincidence because the ground truth representation aligns with the tree structure. I'd like to see some results if the ground truth does not align well.
> >
> > W2: I thank the additional analysis on computational analysis and agree that there is a trade-off between efficiency and performance. What I am actually concerned about is whether the method will maintain good performance when the problem size becomes large (e.g., tree size >= 10 and enumeration becomes extremely time-consuming). Note that for the tree size considered in this submission, it seems that enumerating all possible operator combinations and optimizing weights for each constitutes an alternative solution that is much simpler.
> >
> > Q2: The response addresses my question. But please mention that $\theta$ is a matrix as well as its dimension in the main body, not the appendix. I initially take $\theta$ as a scalar, which raises my original question.
> >
> > Other questions are satisfactory, and I thank the authors for their clarification to them.

---

> > > ### Author Response · Authors · 2025-08-27
> > >
> > > We appreciate your follow-up response. We have highlighted additional revisions in blue.
> > >
> > > Since both you and reviewer 8mN2 raised concerns regarding the tree structure, we have added a new Discussions and Limitations section (page 12) to acknowledge and address these issues.
> > >
> > > W1: Indeed, H-FEX cannot recover the ground-truth expression if the tree structure is incapable of representing it. However, the tree structure does not need to perfectly align with the true expression for H-FEX to succeed. By using identity operators or by learning weights close to zero, larger trees can reduce to subtrees that are functionally equivalent to the target expression. For example, we show that H-FEX recovers the same three-body equation using a slightly different tree structure in the newly added Figure 9 (page 19).
> > >
> > > W2: Trees with 10 or more nodes lead to a combinatorially large operator search space, requiring many more search iterations to reliably identify good operator sequences unless additional methods are used to restrict this space. While parallelization can reduce the cost, it does not scale effectively to very large trees.
> > >
> > > Also, there is inherent variance in the scoring process, so even correct operator combinations can receive poor scores. This variance can arise from randomness in weight initialization and data mini-batching, which lead to varying losses. As a result, a single enumeration of all operator combinations may fail to give high scores to good operator combinations.
> > >
> > > Q2: We now mention that weights $\theta$ may be specified as either scalars or matrices on page 3.

---

### Decision · Action_Editor_BjNq · 2025-09-05

**Recommendation:** Reject

**Additional Comments:**

Since the main shortcomings of the current draft seem to be repairable, I encourage the authors to resubmit their paper once it is properly revised. More concretely, more complicated systems shall be analyzed, and a comparison with other baselines shall be added.

**Audience:**

Yes

**Audience Explanation:**

All reviewers agreed that the topic is potentially interesting for a small community within the readership of TMLR.

**Claims And Evidence:**

No

**Claims Explanation:**

While some reviewers commend the quality of the experiments, some reviewers uttered a major concern that the core contribution of the paper is not convincingly validated. On the one hand, the considered systems were judged as too simple; on the other hand, comparisons with important baselines are said to be missing. Based on this assessment, the reviewers tend to recommend rejection.

**Resubmission Of Major Revision:**

The authors may consider submitting a major revision at a later time.